# Propagation pathways of Indo-Pacific rainfall extremes are modulated by Pacific sea surface temperatures

Felix M. Strnad [1] ✉, Jakob Schlör[1], Ruth Geen [2], Niklas Boers [3,4,5] & Bedartha Goswami [1]

Intraseasonal variation of rainfall extremes within boreal summer in the Indo-Pacific region is driven by the Boreal Summer Intraseasonal Oscillation (BSISO), a quasi-periodic north-eastward movement of convective precipitation from the Indian Ocean to the Western Pacific. Predicting the spatio-temporal location of the BSISO is essential for subseasonal prediction of rainfall extremes but still remains a major challenge due to insufficient understanding of its propagation pathway. Here, using unsupervised machine learning, we characterize how rainfall extremes travel within the region and reveal three distinct propagation modes: north-eastward, eastward-blocked, and quasi-stationary. We show that Pacific sea surface temperatures modulate BSISO propagation − with El Niño-like (La Niña-like) conditions favoring quasi-stationary (eastward-blocked) modes−by changing the background moist static energy via local overturning circulations. Finally, we demonstrate the potential for early warning of rainfall extremes in the region up to four weeks in advance.

The Indo-Asia Pacific region and its population of around 2.5 billion people receive most of its annual rainfall during the monsoon season from June through September (JJAS)[1]. A defining feature of the region is the intraseasonal variation of heavy precipitation and convergent wind circulation[2], which occurs periodically on time scales of around 40 days during boreal summer[3]. Precipitation peaks and troughs are known as "active" and "break" periods[4], the active phase being often marked by widespread extreme rainfall events (EREs)[5]. The intraseasonal timings of active and break periods can leave long-lasting impacts on crop yields and harvest, and suddenly occurring EREs often wreak havoc on rural and urban infrastructures[6].

The Boreal Summer Intraseasonal Oscillation (BSISO) exerts a substantial influence on precipitation dynamics over the oceans and land masses of the Indo-Pacific domain and constitutes a major source of rainfall variability and the occurrence of EREs on intraseasonal time scales[3]. Active phases of the BSISO are initiated in the Indian Ocean triggering a forced Kelvin wave response to the east of the convective anomaly. As the eastward propagating convective system reaches the Maritime Continent, the convection weakens, and moist Rossby waves are emanated, which then move north-westwards toward India. This results in a northwest-southeast tilted band of heavy rainfall that ranges from southern Pakistan in the northwest end to the Philippine Sea and Guam in the southeast. This rainfall band then slowly propagates northward and eastward, finally dissipating over the western Pacific around two weeks later.

Various theories have been put forth to elucidate the BSISO's propagation mechanism. The traditional view suggests that the initiation produces easterly wind flows associated with a slowly eastward propagating convective Kelvin wave[7,8]. The northward propagation is explained via the vertical shear mechanism, which posits that

[1]Machine Learning in Climate Science, University of Tübingen, Tübingen, Germany. [2]School of Geography, Earth and Environmental Sciences, University of Birmingham, Birmingham, UK. [3]School of Engineering & Design, Earth System Modelling, Technical University Munich, Munich, Germany. [4]Potsdam Institute for Climate Impact Research, Potsdam, Germany. [5]Department of Mathematics and Global Systems Institute, University of Exeter, Exeter, UK. ✉e-mail: felix.strnad@uni-tuebingen.de

the background seasonal mean vertical wind shear interacts with upward moving air parcels in the BSISO convective center and, due to the meridional gradient of their vertical velocities, generates cyclonic vorticity and boundary layer convergence to the north of the BSISO cloud band[7,9]. More recently, however, observed fluctuations in sea surface temperatures (SSTs) that appear coherently with BSISO convection[10] have received enhanced attention, emphasizing the role of air-sea interaction. The idea is that the energy source for the propagation is provided by wind-induced surface heat exchange[11] through the feedback processes of the uprising air associated with convection[12]. Based on this mechanism, another compelling explanation has been put forth to account for the northeastward propagation, known as the moisture mode theory[13–15]. It extends the eastward propagation concept of the Madden-Julian Oscillation (MJO)[16], which is mainly observed during boreal winter, to the boreal summer season. Following the initiation of anomalous convection over the tropical Indian Ocean, moistening to the east and drying to the west of the anomalous convection, a typical characteristic of both MJO and BSISO, drive its eastward propagation. It has been explained as a result of the advection of the MJO moisture anomalies by background zonal winds[14], the advection of background moisture by the anomalous flow[17], or the combination of both components[15]. Further studies emphasized the role of the meridional advection caused by the anomalous zonal advection[18] and the coupling between MJO convection and the mean monsoon flow[19] to explain the northward propagation. The theory is supported by empirical evidence derived from observational data[17,20] and has been recently bolstered by advancements in theoretical understanding which incorporate both MJO and BSISO characteristics into a unified framework[21].

Most observational studies concur with the theories and taken together, the combined movement of the eastward and northward propagation characterizes the "canonical" BSISO propagation[2]: A dominant low-frequency mode (30–60 days)[22] in the form of a deep convection zone carrying heavy rainfall emerges in the equatorial Indian Ocean and moves simultaneously eastward and northward, forming a northwest-southeast tilted convection band which, after transgressing the Maritime Continent barrier, progresses further to the Pacific Ocean[2,3,23,24] (exemplified in Figs. S11 and S12). However, not every anomalous convective activity in the Indian Ocean that is associated with the BSISO follows the canonical propagation pathway. Several studies have reported anomalous convective activity that fails to propagate north-eastward and remains stationary in the equatorial Indian Ocean[25,26]. One possible factor modulating the BSISO propagation could be the sea surface temperature (SST) variability associated with the El Niño Southern Oscillation (ENSO), as it is known to affect the rainfall dynamics during the South Asian summer monsoon season, mainly through inducing changes in the Walker circulation[27,28]. But empirical evidence to show clearly that the ENSO influences BSISO propagation is still lacking and the interactions of the Pacific SSTs with the north-eastward propagating convective BSISO system remain poorly understood to date.

Here, we investigate the spatial patterns associated with the propagation of Indo-Asia Pacific rainfall extremes and show that they are clearly linked to different phases of the BSISO. We further address the influence of the SST background state by analyzing the occurrence of synchronous EREs over large spatial areas in the Indo-Pacific domain. We use the fact that the BSISO is a large-scale convective system; thus, BSISO-driven EREs are likely to emerge as spatiotemporally organized weather systems connected via long-range teleconnections[29]. We develop a simple heuristic to identify regions of synchronous BSISO-driven EREs by using the framework of climate networks derived from observational rainfall event data[30]. Our method identifies statistically robust geographical regions that tend to have similar active and break phase timings. BSISO propagation can thus be investigated as the progression of EREs from one region to the next.

Based on these propagation pathways, we cluster them by using an unsupervised spatial clustering method and discover three distinct propagation modes of the BSISO. We further find that the background state in the tropical Pacific does affect BSISO propagation but not its initiation in the equatorial Indian Ocean.

While BSISO's impact on annual monsoonal rainfall has been analyzed thoroughly[3,31–33], the propagation pathways of rainfall extremes linked to the BSISO and the potential influence of the SST background state have received less attention. Previous studies have shown that propagation patterns of convective anomalies during the May-June period exhibit distinct variations compared to those observed from August to October[12] and that ENSO can affect BSISO intensity[34] and propagation over the Maritime Continent. In particular, it was found that the premoistening in the Western Pacific, primarily modulated by ENSO, is influencing the eastward propagation[35]. El Niño-like (La Niña-like) conditions suppress (enhance) the propagation over the Maritime Continent[36]. BSISO activity in the East Asian-western North Pacific region was shown to be influenced by ENSO[37] and the variability in the northward propagation of the BSISO has been related to different cloud hydrometeors[38]. Also, the east-, north- and north-eastward propagation of BSISO-related convection has been investigated, based on predefined propagation directions[18] or on the basis of convective anomalies in the equatorial Indian Ocean[39]. However, these studies do not report any influence of the background SST state on the propagation, and the causes for varying propagation pathways remain unresolved. A mechanistic understanding of the propagation diversity is still lacking, limiting the forecast skill of the BSISO[40] and the ability of numerical models to describe correctly the north-eastward propagation over the Indo-Pacific domain[3,41]. Our work offers a new perspective on BSISO diversity with implications for improving climate model simulations and shows the potential to develop early-warning signals of EREs along the propagation pathway on subseasonal time scales in a prediction period of more than four weeks in advance.

## Results

### Fingerprint of the BSISO on the spatial organization of EREs

In order to explore the BSISO propagation pathway in boreal summer from June through September (JJAS), we first detect its signature in regions with similar active and break phase timings. We thus identify geographical regions where EREs (defined locally as days with rainfall sums above the 90th percentile of wet days) occur synchronously (within up to 10 days) on average over the boreal summer JJAS data period, and whose average ERE timings are distinct from the rest of the study area. The regions identified correspond to "communities" of a climate network constructed by estimating event synchronization[29,42–44] from extreme rainfall event data of the Indo-Asia Pacific domain (illustrated schematically in Fig. 1 and explained in detail in "Methods"). As our community detection model is inherently probabilistic, we repeat the community detection step multiple times and use the distribution of different community detection outputs (Fig. 1c, d) to quantify the membership likelihood of spatial locations of belonging to a particular community (Fig. 1e). The low variances in the shape of the communities (Fig. S4) confirm that the communities are stable manifestations of spatial patterns associated with synchronous EREs.

The community detection reveals six geographical regions, labeled here as the equatorial Indian Ocean (EIO), Bay of Bengal (BoB), Maritime Continent (MC), South Asia (SA), West Pacific (WP), and North India-China (NIC) (Fig. 2a). EIO consists solely of the equatorial and northern Indian Ocean, whereas the BoB region connects India with the Maritime Continent via the Bay of Bengal. The V-shaped form of the BoB region has been reported in modeling studies of the BSISO[7]. SA is a northwest-southeast tilted region connecting the South East Asian Monsoon domain with central India also reported in previous BSISO studies[8,24,39]. WP is located in the North-Western Pacific north of the Maritime Continent and reveals a long south-north shape along the

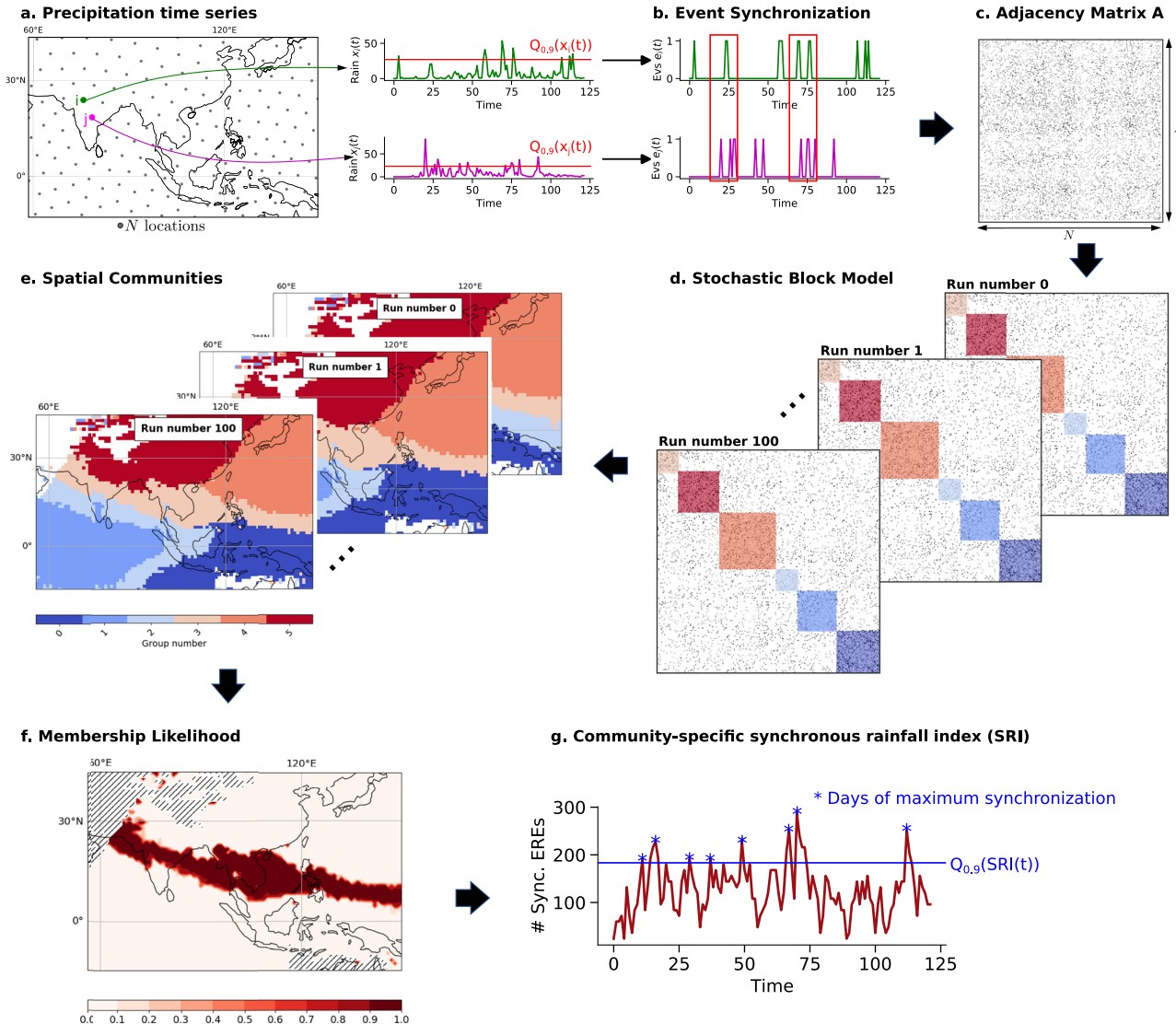

**Fig. 1 | Identification of communities of synchronous Extreme Rainfall Events.**
**a** The dataset comprises a collection of $N$ spatially distributed (depicted as gray circles) rainfall time series (only every 5th location shown). **b** The precipitation time series are transformed into extreme event series, estimated locally by days that exceed the 90th percentile of all wet days. The network is constructed by point-to-point comparison using the event synchronization technique[29,42–44], which quantifies the degree of synchronization between all pairs of single event series ($e_i$, $e_j$) (see "Methods", Eq. (2)). **c** The network is represented by its $N \times N$ adjacency matrix **A**, where $A_{ij} = 1$ if the synchronization between locations $i$ and $j$ is statistically significant (denoted by black dots in **A**). **d** Communities are identified as blocks in **A** by reordering rows and columns using a probabilistic community detection algorithm based on the Stochastic Block Model (SBM, see "Methods"). **e** The network nodes within these blocks correspond to the spatial locations. **f** The membership likelihood describes the probability of a point belonging to a respective community estimated by the overlap of the 100 independent runs. One exemplary community is displayed (other communities: Fig. S4). Hatched regions indicate areas without a sufficiently large number of extreme rainfall events (EREs), and, thus, are excluded from the analysis. **g** The community-specific synchronous rainfall index ($SRI(t)$) counts the number of synchronously occurring EREs per day within a community (see "Methods", Eq. (4)). The time series shown is for illustrative purposes only. The blue stars indicate the local maxima above the 90th percentile of the $SRI(t)$ index, referred to as days of maximum synchronization.

coastline of East Asia including the island of Japan. NIC is solely over land, including the Himalayan Mountains, the Tibetan Plateau, the Ganges Delta, and the Chinese mainland. The regions of synchronization show long spatial extension, e.g., the SA spans over approximately 9000 km from South Pakistan in the west to the West equatorial Pacific in the east. The community structure indicates stable synchronization patterns of EREs in both west-east as well as south-north directions.

**Propagation of EREs.** During boreal summer, EREs propagate along the sequence of the regions EIO → BoB → MC → SA → WP, i.e. from southwest to north-east, in approximately 25 days (Fig. 2b). Such a propagation pathway has also been reported in previous studies[12]. We

find that EREs which occur in EIO are particularly likely to take place in BoB +4 days later (Fig. 2c). In the same way, we see that EREs in BoB are likely to arrive at MC at +4 days later (Fig. 2d), from MC to SA with around +6 days (Fig. 2e) and from SA to WP at approximately +6–11 days later (Fig. 2f). EREs within NIC do not show any significant lagged correlation to the other regions, most likely because it is primarily over land, unlike the other regions. The propagation sequence of EREs along the identified regions is uncovered by using "days of maximum synchronization" within EIO (using the community-specific ERE index, explained in "Methods"), denoted as day 0, and counting the number of EREs in all communities during these and the following 30 days. The maximum time delay between EREs of different communities is estimated by lead-lag correlation analysis. This propagation

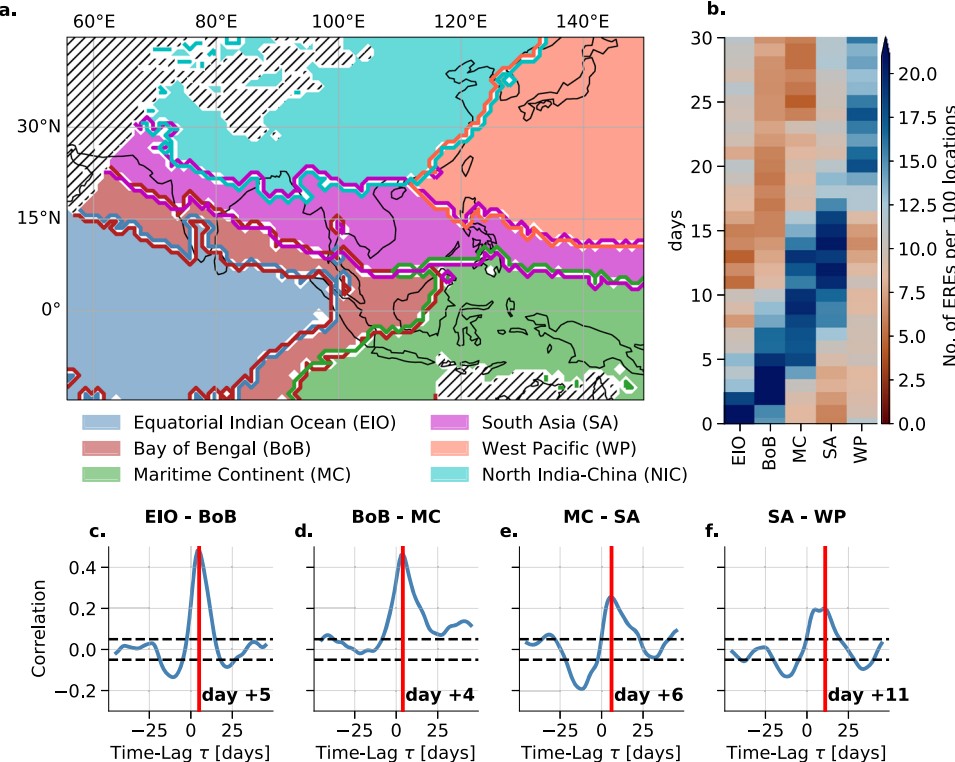

**Fig. 2 | Communities of synchronous extreme rainfall events in the Indo-Pacific domain and propagation characteristics.** The communities are determined using a probabilistic community detection algorithm and overlaps of 100 independent runs. **a** Six regions of synchronous extreme rainfall events (EREs). These are labeled according to their spatial mean position, i.e. equatorial Indian Ocean (EIO), Bay of Bengal (BoB), Maritime Continent (MC), South Asia (SA), Western Pacific (WP), and North India-China (NIC). Hatched areas indicate regions with too little precipitation, which are excluded from the analysis. **b** Temporal evolution of EREs using the most synchronous days within EIO, normalized by the number of grid points per community as day 0. **c–f** Lead-lag correlation analysis between pairs of the synchronous EREs belonging to the communities shown in (a). The time shifts of maximum correlation are denoted by vertical red lines and the respective days are displayed.

scheme can also be observed for specific events (see Supplementary Note 5, Fig. S11, S12).

**BSISO modulates the ERE occurrences.** We find that, except for the NIC, in all regions synchronous EREs are significantly more likely to occur in some particular phases of the BSISO (Fig. 3). We estimate the dependencies between phases of the BSISO (as defined in[23]) and the regions of synchronous EREs (Fig. 2a) using a conditional dependence test interpreted as the conditional probability of synchronous rainfall events subject to (i) phase (ii) active (inactive) BSISO days. As we define days of high synchronization within a community as the top 10% of the community-specific synchronous ERE index, by definition at most 10% of days in JJAS are days of maximum synchronization (dashed lines in Fig. 3). The corresponding null model for a day being a day of high synchronization (P(EREs = 1)) is therefore by construction 10%. For the NIC region, the likelihood of ERES for certain BSISO phases is not considerably different from the null model (Fig. 3f). Therefore, hereafter, we ignore the NIC region and focus on the other five regions (see Supplementary Note 14 for a discussion on the NIC). While an active BSISO increases the likelihood in the communities substantially, inactive days are distributed close to the null model (Fig. 3a–e). The influence of the propagating BSISO on the BoB region (Fig. 3b) also helps to explain the occurrence of the asymmetric V-shaped form (Fig. 2a) resulting from the decreasing zonal wind speed north- and southward of the equator. We observe that the regions that are predominantly oceanic (Fig. 3a–e) show a substantially increased likelihood for certain BSISO phases. The effect of the Maritime Continent barrier is reflected in the comparably lower likelihood for the MC region (Fig. 3c) but it still shows an increased likelihood for BSISO

phase 4 and 5. The dependency of the occurrence of EREs and the BSISO can also be shown by a correlation analysis of the community-specific synchronous ERE indices to the BSISO index (Fig. S18), as well as by a linear regression test (see Supplementary Note 7). We also observe the BSISO characteristic 30-60 day oscillation[2,3,22] in all the community-specific synchronous ERE indices (Fig. S7). The communities MC and WP resemble the 10–20 day oscillation (Fig. S7c, e) which has been reported in that region[45].

## Modes of BSISO propagation determined by Pacific SST background state

The preceding subsection has demonstrated that the BSISO plays a crucial role in shaping the spatial distribution of extreme rainfalls and provides insights into potential propagation pathways (Fig. 2b). However, the signal from the BSISO in the Equatorial Indian Ocean towards the Western Pacific weakens over time (Fig. 2c–f). Therefore, we investigate potential drivers of this diversity in propagation. Since BSISO propagation is clearly accompanied by the ERE progression and the occurrence of regions of highly synchronous rainfalls (Fig. 2), we consider the days of maximum synchronization in the EIO region to be potential BSISO initiation time points. We create for each individual time point two Hovmöller diagrams of outgoing longwave radiation (OLR) anomalies in the zonal and meridional direction to capture both the east- and northward propagation characteristics of the BSISO. The composited Hovmöller diagrams for the initiation time points show discernible patterns of eastward propagation through the Maritime Continent and simultaneous northward propagation, but nevertheless, the clarity of the propagation pattern diminishes approximately 5–10 days post initiation, indicating a degree of variability within the

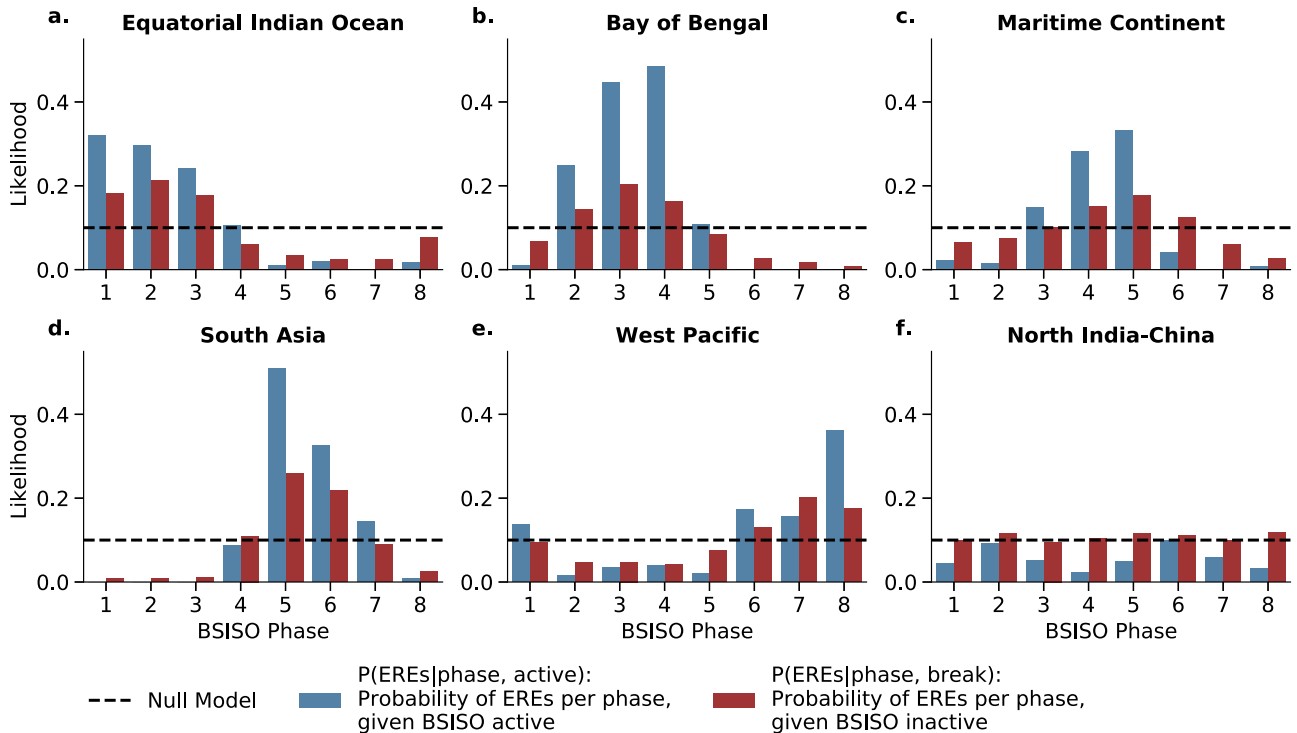

**Fig. 3 | Likelihood of synchronous events for active/inactive Boreal Summer Intraseasonal Oscillation phases.** The likelihood of the occurrence of synchronous events ($P(EREs)$) is analyzed for active (blue) and inactive (red) Boreal Summer Intraseasonal Oscillation (BSISO) phases (as defined here[23]) in the regions of Fig. 2a: **a** equatorial Indian Ocean, **b** Bay of Bengal, **c** Maritime Continent, **d** South Asia, **e** Western Pacific, and **f** North India-China. The dashed line illustrates the likelihood of synchronous events estimated from a null model for synchronous events being randomly selected (i.e. by construction 0.1 as the top 10% of all days are defined as days of maximum synchronization using the community-specific synchronous rainfall extremes index, see "Methods").

propagation pathways (Fig. S8). We thus use the single diagrams for each initiation point in time as input samples to a K-means clustering algorithm (see "Methods"). We obtain three clusters with different propagation features, even though all events initiated in EIO show similar enhanced convection at day 0:

- **Canonical propagation** This propagation cluster consists of 49 distinct Hovmöller diagrams. The convective system reveals a propagation speed in eastward direction of approximately 5.4 m/s (Fig. 4a), and in the northward direction of approx. 1.7 m/s (Fig. 4b). The phase speed in eastward direction is similar to the speed of the related Madden-Julian Oscillation (MJO) propagation[26]. The rainfall anomalies travel from the Indian Ocean towards the Western Pacific, passing the Maritime Continent barrier (Fig. 4a). The transition over the Maritime continent coincides with the temporary decrease of the convection anomalies at around day 10. The east-and northward propagation pattern resembles characteristics described in previous studies[46]. The anomalous wet phase is followed by an anomalously dry phase. The Canonical propagation mode occurs approximately twice as frequently as each non-canonical propagation mode[18]. We confirm that the bipolar pattern between enhanced convection in EIO and dry anomalies at the South Asian mainland and the Western Pacific (Fig. 4a) is characteristic in the eastward propagation[25].

- **Eastward Blocked propagation** This cluster of 28 samples shows a similarly fast northward propagation of 1.4 m/s and a suppressed eastward propagation with a speed of 4 m/s (Fig. 4c) that does not transgress the Maritime Continent barrier (dashed lines in Fig. 4c). Thus, its progression is "eastward blocked". Similar to the Canonical cluster, it progresses to latitudes north of 20° N (Fig. 4b), however, its wet phase is not directly followed by an anomalously dry phase.

- **Quasi-stationary propagation** This mode, consisting of 27 samples, shows a slow propagation that is, however, confined to the Indian Ocean from 50° E to 90° E (Fig. 4e), and a slightly poleward propagation (Fig. 4f) from day −5 to day 10. Further, it is even characterized by anomalously dry conditions in zonal and latitudinal directions besides the enhanced rainfalls in EIO (Fig. 4e, f). Note that keeping the classical definition of the BSISO in mind, these quasi-stationary events should not be called BSISO events as the BSISO is defined by the north-eastward propagation. However, there is literature discussing cases of non-propagating EREs that are still associative with the large-scale modes of variability[25].

**Modulation by SST background state.** We explore how the Pacific SST background is connected to the BSISO propagation. Figure. 5a, c, e shows the background SST anomalies that are associated with the three BSISO modes. The Canonical propagation mode corresponds to conditions without anomalous SSTs in the Pacific (Fig. 5a). A La Niña-like condition, expressed by anomalous cooling in the central Pacific (Fig. 5c), is likely to occur together with the Eastward Blocked Propagation. An El Niño-like condition represented by anomalously warm SSTs in the Pacific Ocean (Fig. 5e) likely occurs together with the Quasi-stationary Propagation. We confirm this connection to ENSO using a conditional dependence test (Fig. S32) showing that the likelihood of a specific BSISO propagation pathway is substantially increased by the respective ENSO state, i.e. Normal conditions favor the Canonical propagation, whereas La Niña- (El Niño-) like conditions favor the Eastward Blocked (Quasi-stationary) mode.

We observe that the column-integrated moist static energy (MSE) is in phase or even leading the spatial distribution of BSISO rainfall anomalies (see Fig. S37) which aligns with the moisture mode theory[17],

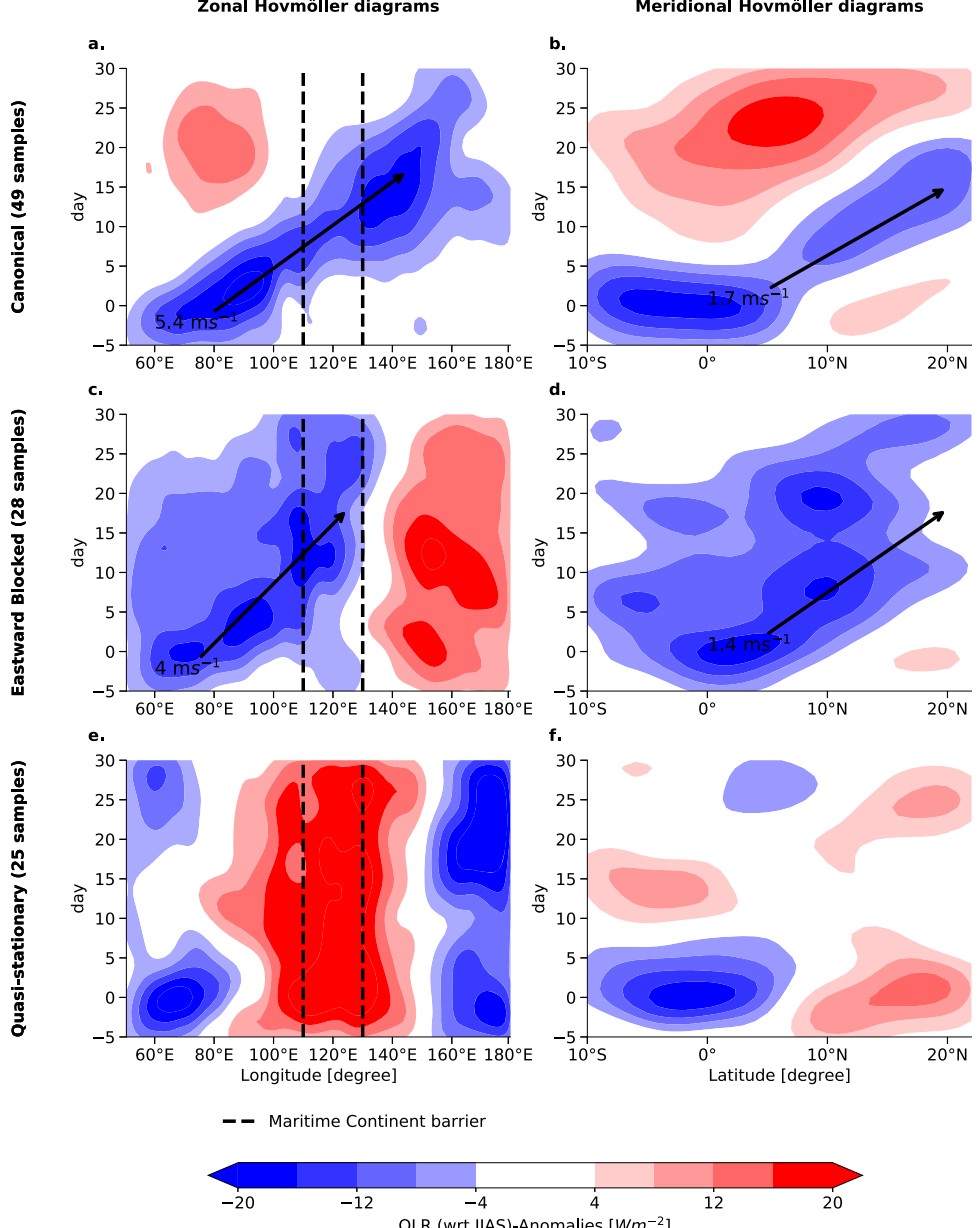

**Fig. 4 | Clustering results of the the different Boreal Summer Intraseasonal Oscillation propagation pathways.** Single Outgoing Longwave Radiation (OLR) Hovmöller diagrams for propagation pathways initiated in the Equatorial Indian Ocean (EIO) community are clustered. The first row shows the propagation for the Canonical mode, the second row the Eastward Blocked mode, and the third row the Quasi-stationary mode. Three different clusters are detected, labeled as Canonical, Eastward Blocked and Quasi-stationary. The first column (**a**, **c**, **e**) shows the composited Hovmöller diagrams in the zonal direction (averaged between [5°S, 5°N]), and the second column (**b**, **d**, **f**) in the meridional direction (averaged between [70°E, 80°E]). All anomalies are computed with respect to the JJAS seasonality. Day 0 describes the days of maximum synchronization within the EIO community (Fig. 2a). The dashed lines mark the area of the Maritime Continent barrier roughly estimated to be from 110° E to 130° E and the black arrows visualize the estimated propagation direction and velocity of the convective system for the propagating modes.

which asserts that meridional and zonal gradients in background MSE enable the propagation of a convective mode such as the BSISO (see Introduction). The Canonical mode (Fig. 5b) exhibits an eastward MSE gradient between the IO and the Western Pacific, crossing the Maritime Continent along with a poleward gradient between the east IO and South East Asia enabling the north-eastward propagation[21]. We observe changes in the background MSE gradient for the Eastward Blocked and the Quasi-stationary mode The Eastward Blocked mode displays a different pattern, characterized by the sign reversal of the zonal gradient around 140° E between the Maritime Continent and the Western Pacific (Fig. 5d), while the poleward gradient persists. Thus for this condition, only a northward propagation is encountered after the

propagation reaches the Maritime Continent. The Quasi-stationary mode, however, exhibits neither a poleward gradient nor an eastward gradient (Fig. 5f) and thus remains trapped in the Indian Ocean. This observation is consistent with observations on the MJO propagation diversity as well[25].

**Interaction of ENSO with BSISO via changes in the overturning circulation.** Changes in the local zonal (i.e. the Walker) and meridional (i.e. the Hadley) overturning circulation (see "Methods") help to understand the changes in background moisture driving the interaction of the BSISO with ENSO. The Pacific Ocean does not show a substantial anomalous zonal wind flow for the Canonical mode (Fig. 6a).

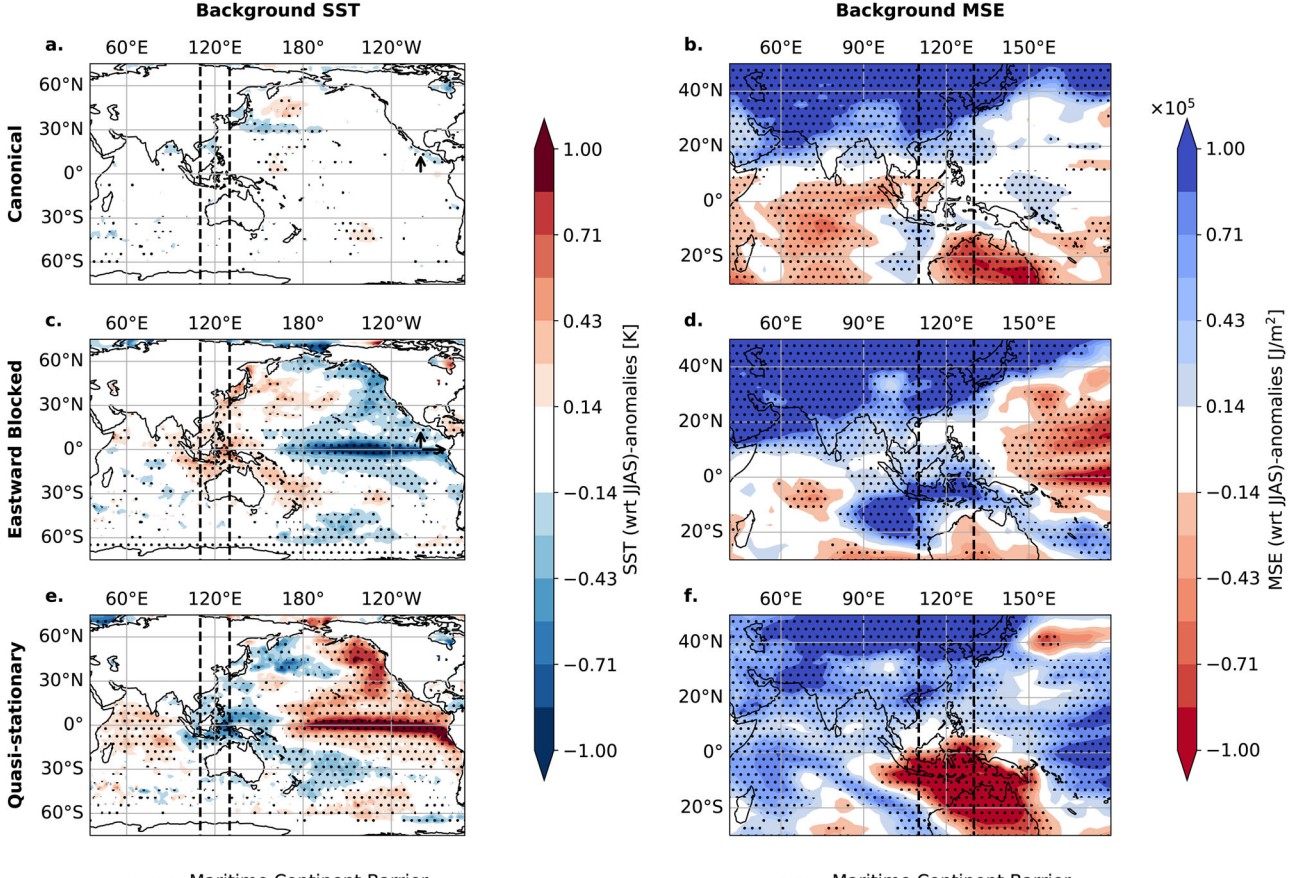

**Fig. 5 | Sea surface temperature and moist static energy background state of the three different Boreal Summer Intraseasonal Oscillation propagation modes.** Background states for sea surface temperature (SST) (**a**, **c**, **e**) and moist static energy (MSE) (**b**, **d**, **f**) for the Canonical (1st row), Eastward Blocked (2nd row), and Quasi-stationary propagation modes (3rd row). Composited anomalies of background column-integrated MSE and SST conditions are shown for days averaged 25–30 days before day 0 (as defined by the days of maximum synchronization), relative to the JJAS climatology. Stipples indicate significance at the 95% confidence level using Student's *t*-test. The dashed lines indicate the region of the Maritime Continent barrier (110° E to 130° E, compare Fig. 4c, e).

The enhanced local zonal overturning circulation in the equatorial Indian Ocean (Fig. 6b) is a consequence of the enhanced convection through the BSISO.

The Eastward Blocked mode reveals an enhanced zonal overturning circulation with the ascending (descending) branch over the Maritime Continent (Fig. 6e, f). The induced anomalous Walker cell over the Pacific Ocean with clockwise circulating air masses connects the circulation in the Indian Ocean with the circulation in the Pacific Ocean. The observed circulation pattern for the Eastward Blocked mode (Fig. 6e) deviates from the conventional La Niña conditions in JJAS (Fig. S34i), featuring a westward displacement of the Walker cell with the updraft of air masses dominated by the BSISO-associated convection (similar to the Canonical mode conditions).

We find two opposing zonal circulations for the Quasi-stationary mode (Fig. 6i). The vertical profile near the Maritime Continent exhibits a descending branch (Fig. 6j) that serves as a barrier, separating the anomalous clockwise zonal circulation in the Indian Ocean from the counterclockwise circulation which is also usually prevalent during El Niño events in the Pacific (Fig. S34j). These descending air masses, in opposition to the convective ascending air masses associated with the BSISO, conceivably suppress the BSISO-associated convection explaining the absence of a zonal MSE gradient (Fig. 5f).

We also observe a strongly enhanced Hadley circulation in the Central Indian Ocean for all three propagation modes (Fig. 6c, g, j) that are the result of the convective anomalies in the EIO. This anomalous overturning circulation pattern in the Indian Ocean is primarily driven by the arising convective anomalies of the BSISO[47]. However, the meridional circulation exhibits strong spatial variations between the different propagation modes (Fig. 6d, h, i), over the Maritime Continent and the Western Pacific. The Eastward Blocked propagation mode shows an elongated convergence corridor over the Western Pacific until the dateline (Fig. 6h) with an opposing circulation to the circulation in the Indian Ocean. The Quasi-stationary mode reveals a bipolar pattern in the region of the Maritime Continent and anomalously ascending air around the equator (Fig. 6l).

**Propagation mechanism.** Taking together the above results, we propose the following mechanism for the three BSISO propagation modes, schematically visualized in Fig. 7. Differences in the ENSO state in the tropical Pacific induce changes in the background MSE conditions over the Maritime Continent via modulation of the local Walker circulation. In all three modes intensified convection is observed in the equatorial Indian Ocean. For the Canonical mode (Fig. 7a) the MSE background condition has a zonal gradient over the Maritime Continent (Fig. 5a) which is induced by the Walker circulation. For the Eastward Blocked mode (Fig. 7b) the La Niña-like conditions trigger enhanced convection over the Maritime Continent (Fig. S35b) and anomalously wet conditions. The ascending air at the Maritime Continent provides an explanation for the eastward blocking (Fig. 4d) at around 120° N since the incoming winds from the Pacific Ocean oppose the eastward propagation of the BSISO convective system. We suggest that this potentially contributes to the observed two opposing MSE gradients preventing the propagation in zonal direction (Fig. 5d). The northward MSE gradient component remains

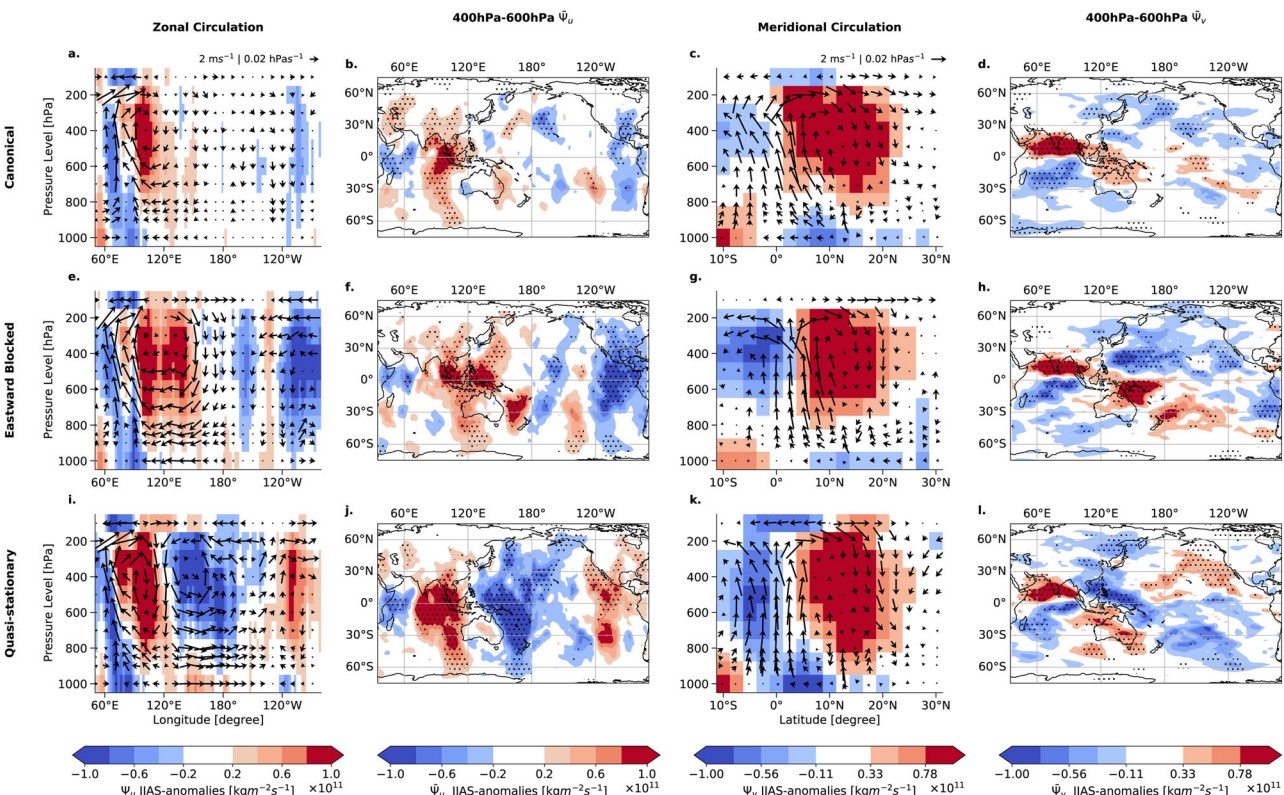

**Fig. 6 | Mass stream function anomalies of overturning circulation for Boreal Summer Intraseasonal Oscillation propagation modes.** For the days 0–5 after initiation in the equatorial Indian Ocean (EIO), the overturning circulation structure is analyzed. The first column (**a, e, i**) shows the composited zonal circulation, meridionally averaged from 0° S to 10° N, the second column (**b, f, j**) the zonally dependent circulation $\bar{\Psi}_u$ averaged between 400 hPa and 600 hPa. The color shading denotes the mass stream function in zonal direction. Red (blue) indicates irrotational lower-level easterlies (westerlies) and upper-level westerlies (easterlies). The third column (**c, g, k**) displays the zonally averaged (between 70°E-80°E) meridional circulation in the Central Indian Ocean. The fourth column (**d, h, l**) depicts the meridionally dependent circulation $\bar{\Psi}_v$ averaged between 400 hPa and 600 hPa. The color shading denotes the mass stream function in the meridional direction. Here, red (blue) indicates irrotational lower-level northerlies (southerlies) and upper-level southerlies (northerlies). The wind fields in the zonal (meridional) circulation plots are estimated using the meridionally (zonally) averaged u (v) components, measured in m/s, and the vertical velocity $w$ in the horizontal direction, measured in hPa/s. For visual clarity, only every 3rd wind arrow is plotted. Stipples denote anomalies that are significant at a 95% confidence level using Student's *t*-test.

unaffected[48]. Contrary, for the Quasi-stationary mode (Fig. 7c) the El Niño-like conditions induce a suppression of convection through the reduced low-level winds over the tropical Pacific leading to anomalously dry conditions over the Maritime Continent. The zonal MSE gradient is therefore already greatly reduced east of the Maritime Continent and the convective system remains over the initiation region in the Equatorial Indian Ocean[49,50].

### Characteristics of the BSISO propagation diversity

Differences in the propagation modes are reflected in differences in terms of the Kelvin- and Rossby wave responses to the anomalous convection in the Equatorial Indian Ocean.

**Eastward propagation.** The Canonical and Eastward Blocked modes resemble a characteristic Kelvin wave signature[15,26,51]. The observed strong easterlies over the Bay of Bengal to the convection center in EIO, expressed by enhanced negative OLR anomalies (Fig. 8a), are previously reported as Kelvin wave responses[51,52] enabling the eastward propagation of the convective cell. Also, the characteristic moistening preceding the eastward-moving convection center is encountered (Fig. S37)[51]. The patterns of the wind fields for the Canonical mode and the Eastward Blocked mode (Fig. 8a, c) are very similar to each other, explaining the similar eastward propagation velocity. The Quasi-stationary propagation mode does not show the characteristic Kelvin wave signature (Fig. 8e) explaining the

quasi-stationary propagation characteristics. We suggest that the descending dry air at the Maritime Continent leads to winds that are opposite to the convective uprising moist air (Fig. 8e). Hence, the Kelvin wave response to the anomalous convection in the EIO fails, so that the deep convection center remains stationary in EIO and vanishes after some days (Fig. 4e).

**Poleward propagation.** We observe the characteristic zonally Rossby wave pattern in the low-level winds[7,8,15,17,19] initiated at the Maritime Continent at ~120° E most clearly for the Canonical mode in Fig. 8b. The meridional wind anomalies reveal a westward drift in northeast-southwest tilted bands. The westward oriented waves occur 10–20 days after initiation consistent with the propagation of EREs (compare Fig. 2b). According to previous literature[7,15,19], the anomalous BSISO circulation near the equator exhibits modified Gill-type responses[53] with a stronger amplitude to the north than to the south of the equator. These are associated with the slanted northwest-southeast BSISO rainfall anomalies near the equator and suppressed convection north of 10$^{circ}$N that we also observe in Fig. 8a, c. Therefore, easterly anomalous winds due to the Rossby wave response (Fig. 8b, e) to BSISO-associated convection generate a meridional dipole-like MSE tendency pattern that drives the northward propagation over the Indian subcontinent at ~70°E (Fig. 8a) in a way as it was already shown in[7]. The Rossby wave pattern for the Eastward Blocked mode (Fig. 8d) is less clearly visible compared to the Canonical mode but starts

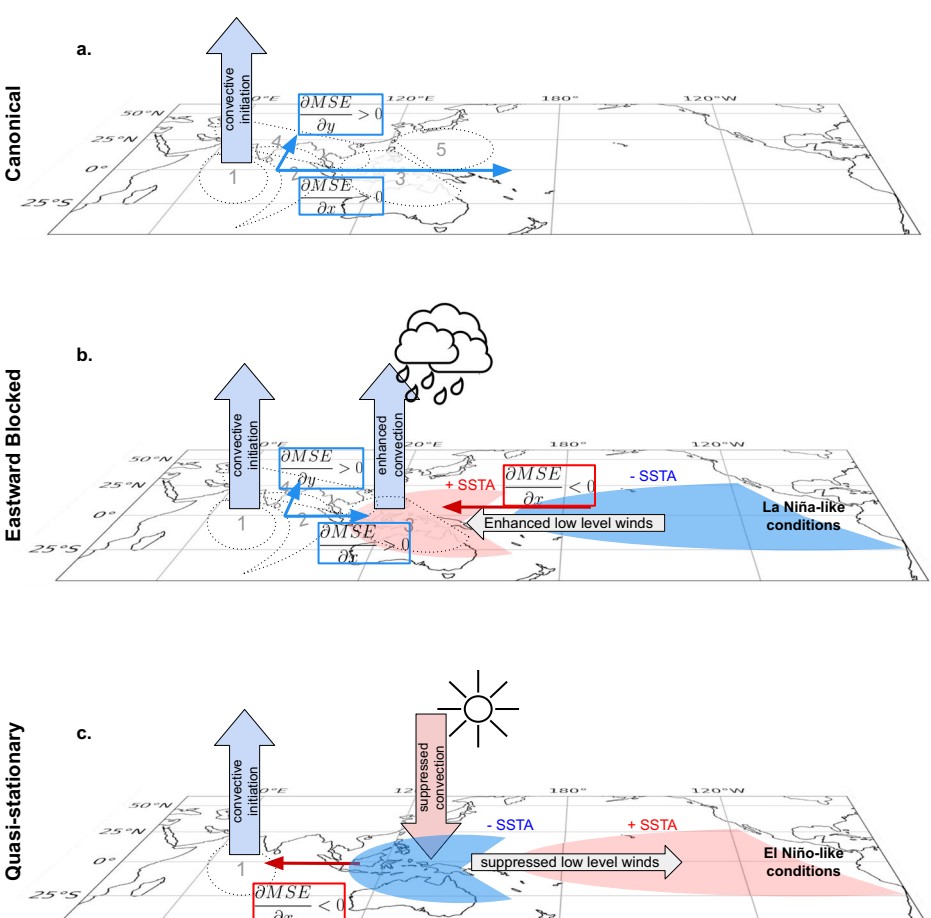

**Fig. 7 | Proposed mechanism for El Niño Southern Oscillation modulation of Boreal Summer Intraseasonal Oscillation propagation modes.** Our proposed mechanism for the three Boreal Summer Intraseasonal Oscillation (BSISO) propagation modes is represented schematically, showing the Canonical propagation (**a**), the Eastward Blocked mode (**b**) and the Quasi-stationary mode (**c**). The increasing (decreasing) gradient of the background moist static energy (MSE) state is visualized in blue (red) boxes in zonal and meridional direction. The respective ENSO condition and its influence on the MSE background state is visualized in the tropical Pacific. The dashed ovals indicate the communities identified (Fig. 2a) and the number denote their propagation in time. Small arrows denote the direction of the background MSE gradients. Big arrows in blue and red indicate the enhanced (suppressed) convection. The rainy clouds (sun) represent anomalously wet (dry) conditions over the Maritime Continent.

occurring at around 10–15 days after initiation in the EIO. The characteristic Rossby wave pattern does not exist in the Quasi-stationary mode (Fig. 8f) which can be explained by the absence of the eastward traveling Kelvin wave and a meridional background MSE gradient.

### Potential for early-warning signals for EREs

The different BSISO propagation modes have a direct consequence on whether or not a given location in the Indo-Pacific region domain experiences an ERE on a given day once a convective anomaly has been initiated in the EIO. For normal conditions without substantial SST anomalies in the tropical Pacific, convective anomalies in the EIO are likely to follow the canonical north-eastward propagation (Fig. 9a and Fig. S13).

For La Niña-like conditions the propagation is trapped in the region of the Maritime continent, and the heavy rainfall remains in India and the South Asian subcontinent for a longer time span (Fig. 9b, and Fig. S14). We further observe that the convective anomalies are more likely to propagate towards higher latitudes in northern India upon the Himalayan foothills (Fig. 9b). If El Niño-like SST conditions coincide with a BSISO initiation in the EIO, our results show that India, the Maritime Continent, and the South Asian mainland are likely to only experience very few convective anomalies (Fig. 9c and Fig. S15). Most of the anomalous convection happens in the central and Western Pacific likely induced by the shifted Walker circulation.

These three propagation modes also translate to propagation of EREs via the identified communities (Fig. 2a), presented in Fig. S16. Using these different propagation modes, it is therefore justified to explore the possibility of early-warning signals (EWS) for EREs that are driven by the BSISO at a time horizon of multiple weeks. We sketch the potential for EWS for the Canonical mode: We estimate the days of maximum synchronization in all regions and calculate the fraction of events that are subsequent to days of maximum synchronization in EIO within a range of three days. In BoB 66% of the days of maximum synchronization occur 4–6 days after days of maximum synchronization in EIO. Subsequently, in the MC region 59% of the days of maximum synchronization are observed 6-9 days later than in the EIO. Similarly, 61% of the events in the SA community events (within 15-18 days and 39% of the events in WP (within 21–24 days) are subsequent to days of maximum synchronization in EIO. Even this simple approach shows the potential for large-scale spatially resolved ERE predictions up to 25 days in advance which is also the target range of the subseasonal to seasonal (S2S) prediction project[54].

### Discussion

The aim of this article has been to reveal the specific BSISO propagation pathways and to improve the mechanistic understanding of the BSISO so that a future potential early-warning system for EREs during boreal summer may be established. We uncovered three dynamically

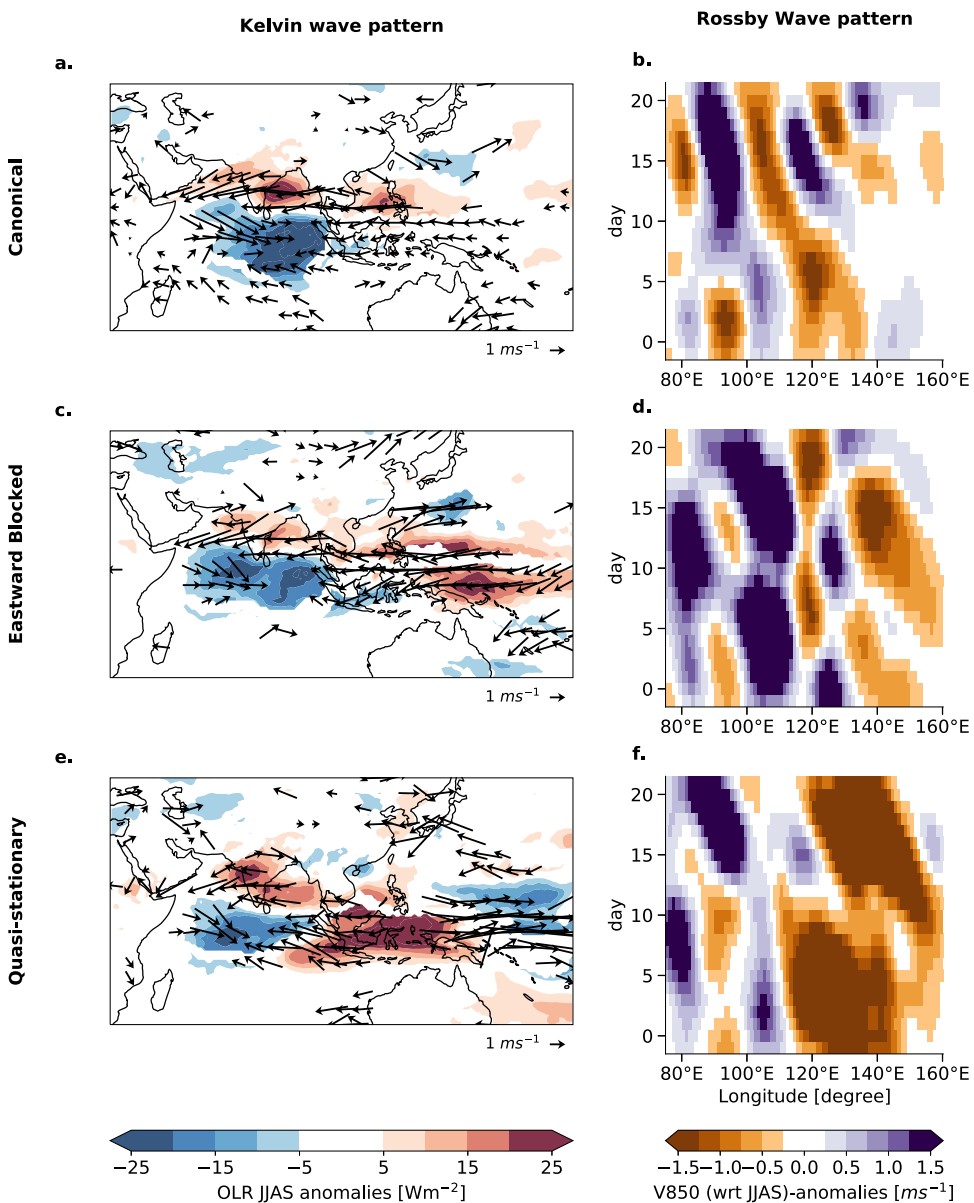

**Fig. 8 | Kelvin and Rossby wave patterns after initiation in the equatorial Indian Ocean.** Different atmospheric conditions as a response to initiation in the equatorial Indian Ocean (EIO) (day 0) are shown for the Canonical (1st row), Eastward Blocked (2nd row), and Quasi-stationary case (3rd row). The first column (**a, c, e**) shows the Kelvin wave pattern by contour lines of outgoing longwave radiation (OLR) and composited wind fields at 850 hPa plotted as arrows averaged for days 2–5 after initiation. Only the statistically significant contour lines and wind field anomaly arrows at 95% confidence level and for visual clarity only every third arrow are shown. The second column (**b, d, f**) shows the corresponding Rossby wave pattern by Hovmöller diagrams of low-level meridional V-winds at 850 hPa. The meridional average is applied for the range of [11°N, 16°N]. Day 0 denotes the initiation in the EIO.

resolved propagation modes of the BSISO that show different spatial manifestations and are strongly influenced by the SST background state of the tropical Pacific Ocean. We argued that understanding these three modes has important implications for the predictability of the occurrence of EREs in the Indo-Pacific region.

We demonstrated that the BSISO is a dominant driver of the spatiotemporal organization of EREs during the boreal summer in the Indo-Pacific region. To uncover the BSISO propagation, we introduced a new approach that combines climate networks based on a non-linear event synchronization measure with a probabilistic network community detection algorithm. Our approach identified macroscale structures of spatially coherent patterns of EREs, involving long-range teleconnections between regions from different parts of the Indo-Pacific domain. The results of our community detection approach are independent of the chosen dataset (see Supplementary Note 8.1) and

remain robust also when using alternative community detection implementations, for instance using the Parallel Louvain Method (PLM) implemented in the NetworkIT packages[55] (see Supplementary Note 8.2). Using a posterior likelihood estimation conditioned on the BSISO index, our representation of spatial BSISO locations revealed a skewed distribution over multiple BSISO phases from the classical definition[23,24]. This confirms the relationship between active and break cycles of monsoon precipitation and the particular phases of the BSISO[3]. Our analysis also provided for the first time a detailed understanding of the spatiotemporal organization of the BSISO-driven rainfall extremes that emerge directly from the data. In this sense, our results present an alternative, impact-focused definition of the BSISO based on ERE data which remains still consistent with the classical definition in terms of atmospheric anomalies. The BSISO is characterized by two modes (10–20 days and 30–60 days) (e.g.[2]). Follow-

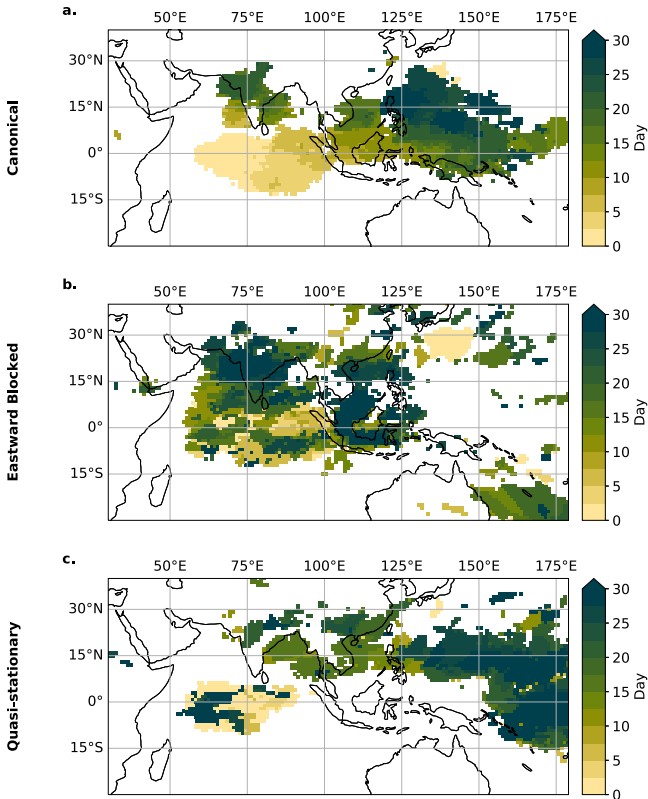

**Fig. 9 | Evolution of anomalous rainfall for the three propagation modes.**
Regions with the highest rainfall intensity for days after maximum synchronization in the equatorial Indian Ocean (EIO) community are shown for the Canonical mode (**a**), the Eastward Blocked mode (**b**), and the Quasi-stationary mode (**c**). Outgoing longwave radiation (OLR) anomalies are computed according to the JJAS climatology. For every day, the statistically significant mean anomalous OLR above the 95th percentile is shown in its respective color of the day. Day 0 denotes the days of maximum synchronization in the EIO community.

up studies could investigate the influence of these modes on the occurrence and propagation of rainfall extremes.

We identified three distinct propagation modes of BSISO-driven EREs: the canonical northeastward propagating mode; the Eastward-blocked mode with continuous propagation only in the northward direction while eastward propagation ceases at the Maritime Continent; and the Quasi-stationary mode. Although the small observational sample size limits the level of statistical significance, our results are robust to randomly chosen initial configurations of the K-means algorithm and subsets of the input data. Note that a recent study reported three different modes of BSISO propagation as well[39], including the "canonical BSISO" propagation mode, and an "Eastward Expansion" mode, which shares some similarities to the Eastward Blocked mode identified in our work. However, the study did not report any modulation of the propagation modes by the background SST state. We contend that the discrepancy may be due to differences in the input to their clustering algorithm, which considered data from May–October and used local minima of an Indian Ocean box-averaged intraseasonal OLR time series to create pentad mean maps of OLR anomalies. Our study focused on the core monsoon season (JJAS) and hence looked at a situation in which the Walker circulation is more shifted to the Western Pacific[56]. In addition, we identified the initialization of BSISO events via regions of highly synchronous EREs ensuring spatially and temporally coherent patterns; and moreover accounted explicitly for zonal and meridional propagation characteristics by using separate Hovmöller diagrams in both these directions.

We reported a plausible mechanism that determines the different propagation modes. Our results provide a new perspective on how ENSO background state interacts with the BSISO on intraseasonal time scales. While the BSISO is known to be marginally correlated with ENSO[3], we showed that the ENSO state strongly influences the BSISO propagation, leading to local variability in rainfall[36,57]. We demonstrated how the coupling of ENSO to the BSISO-driven ERE propagation is mediated by the anomalously dry (moist) background moisture and the cooling (warming) near the Maritime Continent during El Niño (La Niña)-like conditions. We further showed how the interplay between the ENSO-related overturning circulation and the BSISO propagation inhibits the east- and northward propagation of EREs under El Niño-like conditions. This is in agreement with ISM failures observed during El Niño events[28,58]. Conversely, La Niña-like conditions in the tropical Pacific are found to favor the Eastward Blocked mode, such that EREs move only northward and bring extended periods of strong rainfall to the South Asian and Indian mainland as well as to the Maritime Continent. This result further ties in with the longer active spells observed in La Niña years over Indian and South Asian mainland[58,59]. For El Niño conditions the convection is suppressed, whereas for La Niña conditions the anomalous warming at the Maritime Continent leads to opposing winds that prevent the propagation beyond the Maritime Continent. Consequently, the Maritime Continent experiences a weakening (enhancement) of the rainfalls for El Niño (La Niña) conditions corroborating results of ref. 36. Our results thus imply a pronounced role of the Maritime Continent region for the BSISO propagation: The overturning circulation conditions at the Maritime continent are the key to deciding on the most probable propagation type. Our findings, therefore, offer a new perspective on its barrier effect upon existing theories based on the complex topography[60], the high land-sea thermal contrast[61], and the diurnal convection over land[62].

The proposed mechanism of BSISO diversity may provide a framework for understanding why models fail to simulate the BSISO propagation over the Indo-Pacific domain realistically[3,40,41], and thus could offer a new validation scheme for GCM development. In addition, we outlined the potential for medium-range forecasts of EREs and for developing a risk assessment for floods in the South Asian Monsoon domain and along the coast of South East Asia[63,64], on the scale of more than 4 weeks in advance. In comparison, current forecast models of the European Centre for Medium-Range Weather Forecasts (ECMWF) are predicting at a forecast lead time of around 14 days[65]. Although the discussed Early-Warning Signals approach is simplistic, the prediction skill could likely be substantially increased and a location-specific early-warning indicator could be developed by using tools from pattern recognition tasks in machine learning in a similar way as it has been proposed in a recent perspectives paper[66]. However, it should be noted that even though our study identifies the SST variability as a main driving force of the diverse propagation patterns, there are further factors determining the propagation patterns, for example, seasonal differences in the air-sea interaction[12], that need to be considered for an early-warning system.

## Methods

### Data

We use daily precipitation sums for the time period of 1979–2020 from the Multi-Source Weighted-Ensemble Precipitation (MSWEP) V2.2 dataset that merges gauge, satellite, and reanalysis data provided in a resolution of $0.1° × 0.1°$[30]. We used this dataset as it covers a longer time range than other available multi-satellite precipitation products to assure statistical robustness and it has been shown to represent high rainfall quantiles well on both global scales[67] and locally over India[68,69] and South East Asia[70,71]. We restrict our analysis to the tropical Indian Ocean and the South and East Asian Monsoon domain including the Western Pacific (55° E–140° E, 20° S–50° N). We use next-neighbor

interpolation to map the data to a grid of spatially approximately uniformly distributed points employing the Fekete algorithm[72]. The distance between two points corresponds to the spatial distance between two points at the equator of a Gaussian 1° grid, resulting in a total of ≈4700 grid points. We linearly detrend the precipitation time series. The event time series is constructed from 'wet days' only, defined as days with rainfall of at least 1 mm/day. ERE days for a single location are defined as those days where the daily precipitation sum exceeds the 90th percentile of all wet days at that location.

To assess the robustness of our analysis to the choice of dataset, we conducted a comparative investigation using the Tropical Rainfall Measuring Mission (TRMM) dataset[73]. The TRMM dataset was applied to the identical spatial region, interpolated to the equivalent Fekete grid resolution employed for the MSWEP dataset. Our analysis revealed both qualitative and quantitative similarities in the patterns obtained from the TRMM dataset, corroborating our findings derived from the MSWEP dataset (see Figs. S20 and S21).

Further observational datasets used in this study are daily top net thermal radiation, translating to Outgoing Longwave Radiation (OLR), latent heat flux, sea surface temperature, and multi-pressure level variables on 50–1000 hPa of $(u, v)$-wind fields, vertical velocity $w$, and specific humidity $q$ and temperature $T$ taken from the ERA5 Global Reanalysis dataset[74]. The datasets are interpolated to 2.5° × 2.5° grid. The NINO3.4 index[75] was estimated using SST anomaly fields from the ERA5 dataset.

The column-integrated moist static energy (MSE) is a crucial factor in investigating the north-eastward propagation of the BSISO with respect to the moisture mode theory[3,14,21]. It is defined as the sum of sensible heat, latent heat, and potential energy as $MSE = C_p T + L_v q + gz$, where $T$ is temperature, $z$ geopotential height, $q$ specific humidity, $C_p$ the specific heat of air at constant pressure, $g$ the gravitational acceleration, and $L_v$ latent heat of vaporization.

The daily resolved BSISO index by ref. 23 is taken from https://iprc.soest.hawaii.edu/users/kazuyosh/Bimodal_ISO.html (Last Accessed: 10th May 2023). The index is calculated by Principal Component Analysis of OLR for days from May until October. The first two leading principal components (PCs) are used to define the state of the BSISO by the amplitude $A = \sqrt{PC_1^2 + PC_2^2}$, where $A \geq 1$ ($A < 1$) is regarded as active (inactive)[76]. The two-dimensional space spanned by $PC_1$ and $PC_2$ is subdivided into eight equally sized sections that denote the phase of the BSISO. The chosen BSISO index[23] was shown to capture both north- and eastward propagation in a coherent way and is well suited for tracking the BSISO-associated convection[77]. There are further BSISO indices available (e.g.[24,78,79]) but differences in our results remain minor (see Fig. S17 for a comparison).

## Event synchronization-based climate networks

We apply a climate network approach, derived from complex network science. It offers a valuable extension to traditional methods by utilizing the toolset of complex network analysis[80] such as node degree analysis[81] betweenness centrality[82], and network curvature[83], to provide insights into climatic patterns and their interconnections. These network-based metrics provide a non-linear time series analysis of the underlying climate system, allowing conclusions to be drawn that may not be readily accessible through conventional approaches.

Assume a spatiotemporal dataset $X \in \mathbf{R}^{N \times T}$, where $N$ denotes the number of datapoints and $T$ is the number of points in time. The climate network $\mathcal{G}$ is defined as $\mathcal{G} = (V, E)$ where each geographical position of the dataset $x_i(t) \in X$ corresponds to a node $n \in V$ and $E$ is the set of edges. Network edges $e_{ij} \in E$ encode strong statistical dependencies between pairs of time series $x_i(t)$ and $x_j(t)$.

To assess the degree of synchronization between pairs of time series, we use the Event Synchronization algorithm[42]. The number of temporally coinciding events is counted between pairs of event

sequences $\{e_i^m\}_{m=1}^{s_i}$ and $\{e_j^n\}_{n=1}^{s_j}$, where $s_i$ ($s_j$) are the total number of events at location $i$ ($j$), and $e_i^m$ ($e_j^n$) describes the timing of an event in $i$ ($j$). The delay between an event $e_i^m$ in $i$ and an event $e_j^n$ in $j$ is denoted as $d_{ij}^{m,n} = e_i^m - e_j^n$. Defining the set $D_{ij}(e_i^m, e_j^n)$ as the set that contains all four neighboring events of $e_i^m, e_j^n$,

$$D_{i,j}^{m,n} = \left\{ d_{i,i}^{m,m-1}, d_{i,i}^{m,m+1}, d_{j,j}^{n,n-1}, d_{j,j}^{n,n+1}, 2\tau_{max} \right\}, \quad (1)$$

the dynamical delay, $\tau_{ij}^{m,n}$, is defined as half of the minimal waiting time of subsequent events in both time series around event $e_i^m$ and $e_j^n$ and not larger than a predefined maximal value $\tau_{max}$ (Fig. 1b),

$$\tau_{ij}^{m,n} = \frac{1}{2} \min_{\forall d \in D_{ij}^{m,n}} d. \quad (2)$$

It encodes a small deviation between the occurrences, allowing for a time delay between two events. The parameter $\tau_{max}$ separates time scales of ERE synchronization and is set to a maximum delay of $\tau_{max} = 10$ days to ensure both the high- and low-frequency modes of the intraseasonal oscillations are captured. The event synchronization strength $R_{ij}$ between locations $i$ and $j$ is the sum of all synchronous time points between all pairs of event sequences $\{e_i^m\}_{m=1}^{s_i}$ and $\{e_j^n\}_{n=1}^{s_j}$,

$$R_{ij} = \sum_{m=1}^{s_i} \sum_{n=1}^{s_j} S_{i,j}^{m,n} \quad \text{where} \quad S_{i,j}^{m,n} = \begin{cases} 1 & 0 < d_{ij}^{m,n} < \tau_{i,j}^{m,n}, \\ 0 & \text{otherwise} \end{cases} \quad (3)$$

Blocks of consecutive events are counted as one event, placed on the point in time of the first event to avoid the dynamical delay $\tau_{i,j}^{m,n}$ resulting in a value of 1/2, leading to a case where two sequentially occurring events would not be recognized as synchronous.

The adjacency **A** (Fig. 1c) of a network characterizes the interconnections and linkages between nodes, delineating the network's underlying topology. It is a mathematical representation of these connections and captures the presence or absence of links between nodes expressed as a $N \times N$ matrix, where $A_{i,j} = 1$ indicates that events at location $i$ are statistically significantly followed by events at location $j$. We estimate the statistical significance using a null-model test. Our null hypothesis is that an observed $R_{ij}$ value occurs from a pair of purely random event sequences with the same number of events $s_i, s_j$ as in the observed sequences. To encode the null hypothesis, we construct surrogate event sequences $e_i', e_j'$ with $s_i, s_j$ randomly uniformly distributed events. Event series $e_i$ is considered to be significantly synchronous to $e_j$ if their corresponding $R_{ij}$ value is higher than the 95 percentile of $R_{i'j'}$ values obtained using 2000 pairs of surrogate event sequences $e_i', e_j'$. Significant $R_{ij}$ values imply that we place an edge from node $n_i$ to $n_j$ and set $A_{i,j} = 1$.

As the number of comparisons becomes very high even for moderately large datasets (in our case $10^8$ comparisons), there is a non-negligible chance to consider singular pairs of time series as statistically significant, even though their significance is just by coincidence[84]. To avoid such spurious links, we assume that synchronous time series are supposed to be caused by physical mechanisms and thus show spatially coherent patterns[29]. For each spatial location, we rewire its network links randomly 2000 times. We use a Gaussian kernel density estimator (KDE) with the bandwidth selected according to Scott's Rule of Thumb to compute the spatial link distribution of every random sample. A link is only found significant if its regional link distribution (also obtained by a Gaussian KDE) is above the 99.9 percentile.

## Estimating communities within climate networks

Determining macroscale regions of synchronously occurring EREs translates to identifying communities within the network, i.e. groups of nodes in the network that have a much higher number of edges within themselves than to nodes of the rest of the network. To identify communities, we thus need to reorder the rows of the adjacency

matrix **A** such that a clear block structure is obtained (Fig. 1d) which is, in principle a *NP*-hard problem. The problem of identifying communities has been extensively studied in complex network science and many possible solutions have been proposed[85,86]. In our work, we use the Stochastic Block Model (SBM), which essentially formulates a 'block-structure' model of the adjacency matrix in which the edges attached to any given node are determined by two probabilities: a within-block edge probability, and an across-block probability. Thus, a SBM with $k$ blocks would be described by $\frac{k(k+1)}{2}$ probabilities. Typically, the probabilities are then estimated using Bayesian inference techniques. In particular, we use the SBM implementation in the network package `graph_tool`[87,88], as it offers several advantages in terms of speed and efficient data handling. We refer the readers to[87] and[88] for a more detailed explanation of the SBM implementation, but would like to note here that the optimization is guided by the Minimum Description Length (MDL) principle, which favors simpler network structures with equal explanatory power, i.e. it uses the principle of Occam's Razor at its core. The SBM implementation in[87] can thus also fit a hierarchy of SBM's to the observed data and provide a most likely 'optimal' number of communities based on parsimony.

The SBM implementation uses Markov Chain Monte Carlo (MCMC) methods to estimate the posterior likelihoods of model parameters and hence, the inference algorithm is stochastic, meaning that it may produce (slightly) different results at each run[87]. We thus use multiple runs of the model to estimate the uncertainties associated with in assigning nodes to specific communities. In particular, we use a simple heuristic to estimate the posterior likelihood that a geographical location belongs to a particular climate network community: the percent of total runs that a node belongs to a given community is its 'community membership.'

Here, we run the SBM algorithm 100 times on the event synchronization-based climate network, with the constraint that it can have at most 10 communities (as we are interested in large spatial scales). Most of the runs identify 6 communities (with very few exceptions of 5 and 7 communities) with similar spatial shapes (Fig. 1d). Subsequently, by examining the overlap among all 100 SBM runs, we estimate the community membership (Fig. S4).

### Community-specific synchronous ERE index

We introduce the community-specific synchronous ERE index, denoted as $SRI(t)$, to assess for a community its degree of synchronization over time. The index is computed for a particular set of locations $A$, describing one community within the network. For each time step $t$, we count the number of EREs that occur in all event series $e_k$:

$$\text{SRI}_A(t) = \sum_{k \in A} e_k(t). \tag{4}$$

This counting process enables us to quantify the frequency of synchronously occurring EREs within this community $A$ per day. To further pinpoint the points in time of exceptionally strong synchronization, we identify the local maxima in the time series $SRI_A$ that are above the 90th percentile and define these as the "days of maximum synchronization" (blue stars in Fig. 1g).

### Estimation of conditional probabilities

The probability for the occurrence of synchronous rainfall days within a cluster (denoted as $s = 1$) under a condition $a$ is calculated as follows: Assume the set of days with exceptionally synchronous events being $S$ and the set of days that fulfill the condition $a$ being $A$. Then $P(s = 1|a) = \frac{P(s=1,a)}{P(a)} = \frac{||S \cap A||}{||A||}$ describes the conditional probability for synchronous events under condition $a$. Here, $||\cdot||$ denotes set cardinality and $S \cap A$ the intersection of $S$ and $A$. Accordingly, the conditional probability for a second condition $b$ with a set of days $B$ is

computed as:

$$P(s = 1|a, b) = \frac{P(s = 1, a)}{P(a, b)} = \frac{||S \cap A \cap B||}{||A \cap B||}. \tag{5}$$

A corresponding null model is estimated by counting the days of maximum synchronization $||S||$ divided by the total number of days (i.e. in our case ≤0.1). Hence, the upper limit for the null model is $P_{\text{nullmodel}}(s = 1) = 0.1$.

### Clustering of propagation times

The BSISO propagation events we define as a day of maximum synchronization within the region EIO (Fig. 2a and see "Methods") and are denoted as day 0. Consecutive dates by less than 20 days are removed. In total 110 events are considered. We choose the propagation time range to be 5 days before and 30 days after the initiation day 0. The propagation of different synchronous extreme rainfall events is investigated by a *K*-means cluster analysis[89]. To account for the eastward as well as the northward propagation, propagation patterns are analyzed by Hovmöller diagrams of the OLR anomalies along a zonal band averaged between 0°S and 10°N and a meridional band averaged between 70° E and 80° E. OLR is like precipitation an indicator for deep convective activity, but, compared to precipitation, OLR exhibits a considerably smoother pattern, and it offers the advantage of direct measurement without requiring an inverse algorithm (which can introduce errors) as in the case of precipitation[90]. To ignore daily variations and to make the macroscale propagation patterns better distinguishable, we apply a 2D-smoothing Gaussian filter on the Hovmöller diagrams with 5 Pixels as the width of the filter. We use the silhouette coefficient method to determine the optimal number of groups and find that the samples can be best fitted into 3 distinct clusters. The silhouette coefficient indicates how similar a member is to its own cluster. We use it properly to remove outliers that have a silhouette coefficient lower than 0.05 from the cluster analysis. This further reduces the number of input samples by 13 events to 97 events in total.

### Local overturning circulation analysis

In order to assess the relative contributions of the mass fluxes in the troposphere to the pair of meridional and zonal overturning circulations we use the method by[56,91–93]. A Helmholtz decomposition is applied to the global wind field $\mathbf{V} = (u, v)$, to separate the divergent component from the rotational component as $\mathbf{V} = \mathbf{V}_{\text{div}} + \mathbf{V}_{\text{rot}}$. The meridional (zonal) component of the divergent wind $\mathbf{V}_{\text{div}} = (u_{\text{div}}, v_{\text{div}})$ is associated with the north-south (east-west) oriented circulations, commonly known as the Hadley (Walker) cell. The longitudinally dependent meridional circulation is calculated as the mass stream function $\Psi_v$ as a vertical integration over the pressure levels:

$$\Psi_v(\lambda, \phi, p, t) = \frac{2\pi R}{g} cos(\phi) \int_0^p dp' v_{\text{div}}(\lambda, \phi, p', t), \tag{6}$$

where $R$ denotes Earth's radius, $g$ the gravitational constant, $\phi$ latitude, $\lambda$ longitude, $p$ pressure level and $t$ time. The zonal mass stream function $\Psi_u$ is analogously computed as:

$$\Psi_u(\lambda, \phi, p, t) = \frac{2\pi R}{g} \int_0^p dp' u_{\text{div}}(\lambda, \phi, p', t), \tag{7}$$

In our analysis, we use a simplified representation of $\Psi_u$ and $\Psi_v$ by averaging between 400 and 600 hPa.

### Data availability

All data needed to evaluate the conclusions in the paper are present in the paper or the Supplementary Materials. Precipitation data were

taken from the MSWEP dataset (https://www.gloh2o.org/mswep)[30]. Datasets for the composite analysis from 1979 till date were taken from Copernicus Climate Change Service (C3S) (https://cds.climate.copernicus.eu/cdsapp#!/dataset/reanalysis-era5-pressure-levels?tab=overview)[94]. Plots were generated using the Cartopy library[95].

## Code availability

The code for generating and analyzing the networks is made publicly available under ref. 96. The code for reproducing the analysis of the network communities and the spatial clustering described in this paper is publicly available under ref. 97.

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

## Acknowledgements

F.S., J.S., and B.G. acknowledge funding by the Deutsche Forschungsgemeinschaft (DFG, German Research Foundation) under Germany's Excellence Strategy - EXC number 2064/1 - Project number 390727645. F.S. and J.S. thank the International Max Planck Research School for Intelligent Systems (IMPRS-IS) for supporting their PhD program. N.B. acknowledges funding by the Volkswagen foundation, the European Union's Horizon 2020 research and innovation program under grant agreement No. 820970 and under the Marie Sklodowska-Curie grant agreement No. 956170, as well as from the Federal Ministry of Education and Research under grant No. 01LS2001A. We acknowledge support from the Open Access Publication Fund of the University of Tübingen.

## Author contributions

F.S. and B.G. conceived and designed the study. F.S. conducted the analysis. F.S. and B.G. prepared the manuscript. F.S., J.S., R.G., N.B. and B.G. discussed the results and edited the manuscript.

## Funding

## Competing interests
The authors declare no competing interests.
