## [Peer Review File · Nature Communications]

Propagation pathways of Indo-Pacific rainfall extremes are modulated by Pacific sea surface temperaturesREVIEWER COMMENTS

Reviewer #1 (Remarks to the Author):

This study applied the data-driven, complex network approach to identify the propagation pathways of intraseasonal oscillations (ISO) from the Indian Ocean (IO) to western North Pacific (WNP) through the analysis of synchronized extreme rainfall events. As far as I know, this is the first study that applied such data-driven approach to study ISO, and this is certainly innovative. Then through clustering analysis of the propagation profiles the authors have identified the three propagation pathways (canonical, blocked and stationary). While some of these pathways have been studied before individually, this study has established a holistic framework to identify the diversity of pathways.

Another noteworthy result of the study is that there are distinct background sea surface temperature (SST) patterns associated with the three propagation pathways of BSISO, and they are closely related to ENSO. As the authors have pointed out, previously studies have shown that BSISO and ENSO are only marginally correlated (e.g., via BSISO and ENSO-related indices). This study has shown that the different SST patterns would drive different propagation pathways of BSISO, and this result would enable early estimation of extreme precipitation patterns over South Asia to WNP based on forecast of the ENSO phase.

In many studies that applied data-driven approaches to climate systems, physical interpretation has not been well done or even lacking. I appreciate that the authors have paid the effort to analyze the physical processes associated with the various BSISO propagation types, such as through the Walker and Hadley circulation analysis and based on the Kelvin-Rossby wave responses, which are the theories that the existing literature is based on. Overall, I think this manuscript should be considered for publication, after some clarifications, better linkages to the previous literature and further improvements in interpreting the methodology, especially that associated with the complex network construction using the event synchronization (ES) technique. The following are my specific comments.

1. BSISO has a high-frequency (10-20 day) mode and a low-frequency (\sim 25-70 day) mode, the latter resembles the Madden-Julian Oscillation (MJO). The high-frequency mode has propagation over the IO (Wen et al. 2010, doi:/10.1175/2009JAS3105.1) and also distinct northward propagation over the WNP (Gao et al. 2016, doi10.1007/s00382-016-3045-3). The low-frequency mode also has distinct propagation over the WNP (Lin 2013, doi:/ 10.1175/MWR-D-12-00087.1). In the ES technique applied in this study, the maximum time delay between events was 10 days. I think by this time delay, both the high-frequency and low-frequency modes of propagation have been included. There is nothing wrong in the current application of ES. However, I think the authors should provide such background in our current understanding of BSISO. There is also good potential in follow-up studies to tune the ES techniques such that the two modes of BSISO can be separately analyzed.

2. Propagation patterns of BSISO (on MJO timescale) have been analyzed in details in Kemball-Cook and Wang (2001, doi:/10.1175/1520-0442(2001)014<2923:EWAASI>2.0.CO;2). See the schematics in their Figure 11, which nicely resembles the identified network communities in this study. They have identified that the propagation patterns during MJ and ASO are not the same. The current study analyzed the JJAS period, which can be considered a mean boreal summer propagation pattern. The sub-seasonal variation identified in Kemball-Cook and Wang may be discussed in the manuscript.

3. Also in Kemball-Cook and Wang (2001), air-sea interaction has been emphasized as a mechanism for propagation of convective anomalies, besides the Kelvin-Rossby wave response. Air-sea interaction is through the feedback processes from updraft/downdraft associated with convection to SST/skin temperature, downward shortwave radiation, latent heat flux and then moist static energy. Such mechanism may be briefly discussed in the manuscript.

4. In regard to the network communities identified from the ES technique, it is nice to see that they resemble the propagation patterns of convection during BSISO as what we have learnt from

previous studies, such as the origin of convection in west equatorial IO and the V-shape BoB communities associated with the northward and southward propagation. However, there must be some distinct features of the extreme rainfall events within a community such that they are highly connected (i.e., synchronized). In many previous studies that applied the ES techniques, network communities are often associated with different rainfall systems, such as tropical cyclone and frontal systems. In studies that focused on monsoon precipitation, network communities are often associated with rainfall over land and that over ocean (e.g., the separation between the NIC and WP communities here), where rainfall characteristics are not the same (e.g., the development of more or less mesoscale convective systems). Some kinds of network metrics are often applied to "explain" the communities, such as degree centrality, betweenness, local clustering coefficient and transitivity. Have the authors examined network metrics like these and able to identify distinct characteristics in the communities? They may provide us clues about characteristic changes in the convective anomalies during the propagation.

5. The SST pattern associated with the eastward blocked propagation pathway is certainly a La Niña-like pattern (Figure 5b). However, the accompanied zonal and vertical circulation (Figure 6e) is not like a strengthened Walker circulation during a usual La Niña but a more isolated clockwise cell. I think the composite in Figure 6e has been dominated by the updraft and downdraft associated with the BSISO convection.

Reviewer #2 (Remarks to the Author):

The authors study the influence of BSISO on the rainfall extremes in the South Asian Summer Monsoon region. They found three distinct BSISO propagation modes: north-eastward, eastward-blocked, and stationary. It is shown that Pacific sea surface temperatures can modulate these modes - El Niño-like conditions favors the stationary, while La Niña-like conditions favors eastward-blocked. These findings, if validated, can have significant implications for early warning of extreme rainfall events. The manuscript is overall concisely written, but sometimes difficult to read due to technicality. I feel that the manuscript has great potential, but needs to be improved. The proposed mechanism of BSISO propagation models is not convincing yet, and further analysis would be very helpful. I recommend major revision for publication.

Main comments:

1. The authors use the method of climate network, which is rarely used in weather and climate sciences. As it is written in the present form, it's difficult to understand what the method is really doing, and why the authors make the choice using this method. For example, what is the goal of this algorithm? What's the cost function? What's the underlying probabilistic assumption/model? It appears that estimating adjacent matrix is of central importance. How to interpret it in Fig. 1c? What's the horizontal/vertical dimension? (Are those grid points? if so, how many?) Is the it judged by density of dots? What parameters the MCMC method is used to train? The algorithm is opaque to readers in atmospheric sciences in the present manuscript. It should be much improved for publication. I also suggest give some examples (some specific ERE events) for illustrative purpose. The ERE index is the outcome of this method. It also needs clarified.

2. The authors study the relationship between the BSISO diversity and ENSO. In Page 4, Fig. 4, Fig. 9, the authors refer to the third mode is Fig. 4 as "stationary". There is some indication that this mode has some convective signals in the western Pacific (the plotting domain only goes to 140E, please expand this further east, as Fig. 9). Fig. 9 also shows that there are substantial signals in the western Pacific region after 20 days, indicating eastward propagation (even though it appears not continuous). It appears that this is not stationary mode, but jumps across the MC region and reboots in the western Pacific.

3. The authors attribute northward propagation of the BSISO to vertical wind shear. But this is not the only mechanism. Recent observation analysis (Jiang et al 2018) and theory (Wang and Sobel 2022) further show that background moisture is crucial for the propagation of the BSISO in the Indian Ocean, where seasonal mean column moisture increase northward and eastward. The

theoretical study shows that the BSISO can have both eastward and poleward propagation, and the phase tilts northwestward – southeastward in the presence of poleward gradient of moisture, both consistent with observed phase tilt of the BSISO in the north hemisphere. Given these prior studies emphasizing important roles of moisture for the BSISO, I suggest the authors also examine moisture in different ENSO years (El Nino, La Nina etc) to see if the moisture gradients are significantly different in these years, leading to different BSISO modes, i.e., propagation, or blocking, or jumping.

Reference:

Jiang, X., A. Adames, M. Zhao, D. Waliser, and E. Maloney, 2018: A unified moisture mode framework for seasonality of the Madden–Julian oscillation. *J. Climate*, 31, 4215–4224, <https://doi.org/10.1175/JCLI-D-17-0671.1>.

Wang, S., A.H. Sobel. 2022: A unified moisture mode theory for the Madden Julian Oscillation and the Boreal Summer Intraseasonal Oscillation. *J. Climate*. 35(4), 1267-1291. <https://doi.org/10.1175/JCLI-D-21-0361.1>

Specific comments:

P1. Abstract. The author emphasize they focus on the region of South Asian Summer Monsoon. Nevertheless, the region the authors study extends far beyond SASM (e.g., Fig. 3a). It covers most east Asia, Maritime Continent, and vast western Pacific Ocean. I would suggest use another name.

P3, what is null model? Why it is by construction 10%?

P4. Are the ERE index used to compute the lag correlation in Figure 4c? The ERE index needs to be shown and checked/compared with other BSISO index.

P6, Figure 4, the unit for OLR anomalies is W/m². Figure 4 shows that OLR anomalies are in the range of -60-60 x10³ W/m². This is a huge number for OLR and it is not in a reasonable range.

P11, Figure 8 shows mean vertical shear, dw/dy , and relative vorticity. The later two variables look like the BSISO structure. That is, it's a result of BSISO, not the cause. Thus it is hard to use this to explain the different propagation of the BSISO modes.

P13, 2nd paragraph in the Data section.

Synchronicity: not defined

delete "according to"

P13: The BSISO index used in the manuscript has difficulty capturing northward propagation according to lag correlation analysis (see reference 50). Please use another one in ref 48 or 49).

P15, 3rd paragraph: "OLR is used instead of precipitation because it directly indicates deep convection in the tropics." This is not a valid argument. OLR is an indicator of convection, but it's no better than precipitation. A better argument could be that OLR is lot smoother than precipitation, and OLR is direct measurement and it does not need inverse algorithm (which could have errors) like precipitation.

P15, last paragraph: where the Helmholtz decomposition is applied? Global domain or only tropics?

P13, Data section: The MSWEP dataset is a derived product. It needs to be calibrated with surface observation. Do you have reference for this?

Figure S1 caption: "An Extreme Rainfall Event (ERE) is defined as a day with more than 1 mm/day of precipitation." This doesn't sound right.

Reviewer #3 (Remarks to the Author):

This article examines South Asian extreme rainfall and propagation of the boreal summer intraseasonal oscillation (BSISO). Exploiting machine learning, spatial patterns and propagation pattern of precipitation are identified. The propagation mechanisms and the role of ENSO are explored. I find the results inspiring, but I suppose it is not acceptable until the following comments are addressed.

Major Comments:

1. Data. Could you only use data in boreal summer? For two reasons. (1) The study seeks to address South Asian extreme rainfall. (2) Community "BoB" bends back in the Southern Hemisphere, which I assume is dominated by MJOs in the boreal winter. The results (Figures 5-8) could be cleaner if you focus on the boreal summer. The results are inspiring, but I would love to hold back a bit till the author eliminate seasonality in the analysis.

2. The first three figures are disconnected with the rest of the figures. Community detection is a novel method to study the BSISO and indeed, it yields similar pattern as that from the more conventional EOF analysis. But as I thought the authors would complete the story in the newly defined "community" space, the rest of the article returns to conventional K-mean clustering in the conventional Hovmöller diagram. I am not suggesting conventional methods are worse choices – I just do not understand why community detection is necessary for this study.

Minor Comments:

1. How is the number of communities determined? Is it sensitive to the chosen community detection algorithm?

Felix Strnad
University of Tübingen
Cluster of Excellence 'Machine Learning'
Maria-von-Linden-Strasse 6,
D-72076 Tübingen
email: felix.strnad@uni-tuebingen.de

Tübingen, July 6, 2023

Response letter for “Propagation pathways of Indo-Pacific rainfall extremes are modulated by Pacific sea surface temperatures”

Point-by-point responses to reviewer #1

This study applied the data-driven, complex network approach to identify the propagation pathways of intraseasonal oscillations (ISO) from the Indian Ocean (IO) to the western North Pacific (WNP) through the analysis of synchronized extreme rainfall events. As far as I know, this is the first study that applied such a data-driven approach to studying ISO, and this is certainly innovative. Then through clustering analysis of the propagation profiles, the authors have identified the three propagation pathways (canonical, blocked, and stationary). While some of these pathways have been studied individually, this study has established a holistic framework to identify the diversity of pathways. Another noteworthy result of the study is that there are distinct background sea surface temperature (SST) patterns associated with the three propagation pathways of BSISO, and they are closely related to ENSO. As the authors have pointed out, previous studies have shown that BSISO and ENSO are only marginally correlated (e.g., via BSISO and ENSO-related indices). This study has shown that the different SST patterns would drive different propagation pathways of BSISO, and this result would enable early estimation of extreme precipitation patterns over South Asia to WNP based on the forecast of the ENSO phase. In many studies that applied data-driven approaches to climate systems, physical interpretation has not been well done or even lacking. I appreciate that the authors have paid an effort to analyze the physical processes associated with the various BSISO propagation types, such as through the Walker and Hadley circulation analysis and based on the Kelvin-Rossby wave responses, which are the theories that the existing literature is based on. Overall, I think this manuscript should be considered for publication, after some clarifications, better linkages to the previous literature, and further improvements in interpreting the methodology, especially that associated with the complex network construction using the event synchronization (ES) technique. The following are my specific comments.

We would like to thank the reviewer for the careful evaluation of our manuscript and their thoughtful comments. We are happy that the results of this study are considered noteworthy and appreciate that the effort invested in identifying a physical interpretation has been recognized. We agree on the need for a better linkage to the literature and are grateful for the suggestions that the reviewer provided. We also agree that the methodology should provide a better intuition with respect to the applied climate network. Thus, large parts of the methods section have been fully rewritten and the introduction and discussion section have been extended to incorporate the comments thoroughly.

Specific comments:

1. *BSISO has a high-frequency (10 – 20 day) mode and a low-frequency (~ 25 –70 day) mode, the latter resembles the Madden-Julian Oscillation (MJO). The high-frequency mode has propagation over the IO (Wen et al., 2010), and also distinct northward propagation over the WNP (Gao et al., 2016). The low-frequency mode also has distinct propagation over the WNP (Lin, 2013). In the ES technique applied in this study, the maximum time delay between events was 10 days. I think by this time delay, both the high-frequency and low-frequency modes of propagation have been included. There is nothing wrong with the current application of ES. However, I think the authors should provide such background in our current understanding of BSISO. There is also good potential in follow-up studies to tune the ES techniques such that the two modes of BSISO can be separately analyzed.*

We thank the reviewer for highlighting this important aspect. We agree that more information on the frequency of the BSISO should be provided, which has been done in the revised version of the manuscript. We have applied a power spectrum analysis of the synchronous event indices of the different communities (see SI Fig. S7). It shows clearly significant peaks for periods between 30-60 days for the 5 BSISO communities considered in this study. Moreover, these frequency peaks are not present in the power spectrum of the synchronous index of the NIC community which is consistent with our further analysis that the NIC community is not BSISO dominated. The synchronous rainfall index here resembles nicely the power spectra of the BSISO indices (Kikuchi et al., 2012). We find, as expected by the reviewer significant peaks for 10-20 days at the region of the WP and MC. We discuss this observation in the SI and added the following reference to it in the results section: “*The communities MC and WP resemble the 10-20 day oscillation which has been reported in that region (Gao et al., 2016).*” Further, in the introduction we provide background information by adding the phrase “*A defining feature of the South Asian Summer Monsoon (SASM) is the intraseasonal variation of heavy precipitation and convergent wind circulation, which occurs periodically on time scales of around 40 days Kikuchi (2021).*”. In the discussion section, we also incorporated the reviewer’s valid suggestions to mention the potential for analyzing the two modes in follow-up studies: “*Follow-up studies may now more in-depth investigate the potential influence of the two modes on the propagation extremes (10-20 days and 30-60 days) of the BSISO (Wang, 2018).*” In the methodology, we now justify the choice of $\tau_{\max} = 10$ days by the following phrase: “*The parameter τ_{\max} separates timescales of ERE synchronization and is set to a maximum delay of $\tau_{\max} = 10$ days to ensure capturing both the high-frequency and low-frequency modes of the intraseasonal oscillations*”.

2. *Propagation patterns of BSISO (on MJO timescale) have been analyzed in detail in Kembball-Cook and Wang (2001). See the schematics in Figure 11 which nicely resembles the identified network communities in this study. They have identified that the propagation patterns during MJ and ASO are not the same. The current study analyzed the JJAS period, which can be considered a mean boreal summer propagation pattern. The sub-seasonal variation identified in Kembball-Cook and Wang (2001) may be discussed in the manuscript.*

We thank the reviewer for pointing us toward the work of Kemball-Cook and Wang (2001) which indeed fits very well with our study. We have added the following phrase in the introduction: *Previous studies that have worked on this problem have shown that propagation patterns of convective anomalies during the May-June period exhibit distinct variations compared to those observed from August to October (Kemball-Cook and Wang, 2001).* To test whether our identified pathways show a preference for a specific month or are a mean boreal summer pattern, we created histograms of the occurrence of EREs per month and week (see sec. SI IV). We observe that in the JJAS period, the occurrence of the most synchronous days per community is distributed roughly uniformly (see Fig. S8, S9). Subseasonal variations of the propagation pathways were therefore not identified in this study and the signal of ENSO turned out to be the dominating force behind the diversity in the propagation pathways detectable by our clustering approach. We agree that it would be interesting to study the intraseasonal variability of our identified propagation modes, however, this goes beyond the scope of this project and is the subject of ongoing research in our group. In the discussion section we have added the phrase *“However, it should be noted that even though this study identifies the SST variability as a main driving force of the diverse propagation patterns, there are further factors determining the propagation patterns as for example sub-seasonal differences (Kemball-Cook and Wang, 2001) that need to be considered for an early warning system”.*

3. *Also in (Kemball-Cook and Wang, 2001), air-sea interaction has been emphasized as a mechanism for the propagation of convective anomalies, besides the Kelvin-Rossby wave response. Air-sea interaction is through the feedback processes from updraft/downdraft associated with convection to SST/skin temperature, downward shortwave radiation, latent heat flux, and then moist static energy. Such mechanism may be briefly discussed in the manuscript.*

Thank you for pointing us towards the need to better discuss the air-sea interaction. It is one basic component on which the so-called moisture mode theory is built (Sobel and Maloney, 2012; Adames and Kim, 2016). We have added the following phrases to the introduction: *“More recently, the observed fluctuation in sea surface temperature (SST) which appears coherently with BSISO convection (Sengupta et al., 2001) has received enhanced attention emphasizing the role of air-sea interaction. The idea is that the energy source for the propagation is provided by wind-induced surface heat exchange (Sobel et al., 2008) through the feedback processes of the uprising air associated with convection (Kemball-Cook and Wang, 2001).”* Inspired by this theory, we found that the east- and poleward increasing MSE gradient is different for the different propagation types (new Fig. 5 b,d,f). We have also included a new schematic (Fig. 7) to better guide the reader’s intuition on the proposed mechanism. As suggested by the reviewer, we also analyzed the influence of latent feedbacks associated with the updraft/downdraft of BSISO-associated convection. These clearly show the signature of the BSISO-related convection shown in the plots of the downward shortwave radiation, latent heat flux, and moist static energy in the section on the moisture mode theory in SI Fig. S34 and Fig. S35. This important concern was also raised by Review 2. Therefore, we would like to further refer the reviewer to our answer to point 3 of reviewer 2 below.

4. *In regard to the network communities identified from the ES technique, it is nice to*

see that they resemble the propagation patterns of convection during BSISO as what we have learned from previous studies, such as the origin of convection in the west equatorial IO and the V-shape BoB communities associated with the northward and southward propagation. However, there must be some distinct features of the extreme rainfall events within a community such that they are highly connected (i.e., synchronized). In many previous studies that applied the ES techniques, network communities are often associated with different rainfall systems, such as tropical cyclones and frontal systems. In studies that focused on monsoon precipitation, network communities are often associated with rainfall over land and over the ocean (e.g., the separation between the NIC and WP communities here), where rainfall characteristics are not the same (e.g., the development of more or less mesoscale convective systems). Some kinds of network metrics are often applied to “explain” the communities, such as degree, betweenness centrality, local clustering coefficient, and transitivity. Have the authors examined network metrics like these and were able to identify distinct characteristics in the communities? They may provide us with clues about characteristic changes in the convective anomalies during propagation.

We are thankful to the reviewer for highlighting this important aspect. In general, we agree that the dynamics of different mesoscale systems and the boreal summer atmospheric circulation is reflected by the network structure. Note that it is very context-specific how the ES algorithm is applied in other studies. I.e., both the spatial range and the time range differ when, for instance, analyzing tropical cyclones (Gupta et al., 2021) compared with long-range teleconnections (Boers et al., 2019). We thank the reviewer for the idea of analyzing the communities by different network metrics. This analysis for node degree, the local clustering coefficient, curvature, and betweenness centrality is shown in SI IX, Fig.S24-S28. It is true that the separation between land and sea might be a possible explanation for the different clustering coefficient distributions for NIC and the other communities that are mainly over the ocean (SI Fig. S28). However, in our interpretation, we do not find any other characteristics that allow to provide a good explanation for the shape of the communities. Therefore, it seems to be the case that the most distinguishing feature is the difference in the local edge densities which is also what the SBM optimizes when identifying the community structure. Similarly, we have applied an analysis for boundary corrected (Rheinwalt et al., 2012) measures node degree, betweenness centrality, and clustering coefficient on the same domain as for the climate network approach (SI IX, Fig.S22-S28) but also this analysis did not bring the hoped-for better explanations.

5. *The SST pattern associated with the eastward blocked propagation pathway is certainly a La Niña-like pattern (Figure 5b). However, the accompanied zonal and vertical circulation (Figure 6e) is not like a strengthened Walker circulation during a usual La Niña but a more isolated clockwise cell. I think the composite in Figure 6e has been dominated by the updraft and downdraft associated with the BSISO convection.*

We appreciate the reviewer’s observation regarding the SST pattern associated with the eastward blocked propagation pathway (Figure 5b). In SI Fig. S28, we plotted the mean JJAS circulation structures for El Niño (La Niña) like patterns (independent of the BSISO). It is important to note that the circulation patterns we analyzed in our study primarily pertain to the boreal summer JJAS season. In contrast, the classical El Niño (La Niña) circulation patterns often focus on winter circulation over the Pacific. This distinction in seasonal context might also contribute to the differences observed

between the SST pattern and the circulation cells in that in boreal summer the structure of the Walker cell in boreal summer is already more shifted towards the west Pacific (Schwendike et al., 2014). Still, we acknowledge that the accompanied zonal and vertical circulation (Figure 6e) does exhibit differences to the characteristic strengthened Walker circulation for a classical La Nina like condition in JJAS (SI Fig. S28 i) in that it is even more shifted towards the west. We agree with the reviewer’s insight that the increased updraft over the Maritime Continent is dominated by the BSISO. We also agree that our earlier version could have emphasized the differences in the Walker cell better and reported the influence of the BSISO associated convection. We have made this now more explicit in the revised manuscript: “*The observed circulation pattern for the Eastward Blocked mode (Fig. 5 e) deviates from the conventional La Niña condition in JJAS (Fig. S28 i), featuring a westward displacement of the Walker cell with the updraft of air Masses are dominated by the BSISO associated convection (similar to the Canonical mode conditions).*” Exploring the underlying mechanisms and factors that lead to this shift and how this might be connected to the updraft (downdraft) associated with the BSISO convection is subject to current research in our group but goes beyond the scope of this article.

Point-by-point responses to reviewer #2

The authors study the influence of BSISO on the rainfall extremes in the South Asian Summer Monsoon region. They found three distinct BSISO propagation modes: north-eastward, eastward-blocked, and stationary. It is shown that Pacific sea surface temperatures can modulate these modes - El Niño-like conditions favor the stationary, while La Niña-like conditions favor eastward-blocked. These findings, if validated, can have significant implications for early warning of extreme rainfall events. The manuscript is overall concisely written, but sometimes difficult to read due to technicality. I feel that the manuscript has great potential, but needs to be improved. The proposed mechanism of BSISO propagation models is not convincing yet, and further analysis would be very helpful. I recommend major revision for publication.

We first want to thank the reviewer for their time and their very helpful and detailed comments. These comments have helped us to better interpret the results and we believe that our manuscript has benefited a lot. We apologize for the technicality and the lack of reference to current BSISO propagation models. Therefore, we have rewritten most parts of the methodology, introduced a new schematic to better visualize our proposed mechanism and extended the results section by interpreting our results with respect to the moisture mode theory.

Please find below our answers to the main and specific comments.

Main comments:

1. *The authors use the method of climate network, which is rarely used in weather and climate sciences. As it is written in the present form, it's difficult to understand what the method is really doing, and why the authors make the choice using this method. For example, what is the goal of this algorithm? What's the cost function? What's the underlying probabilistic assumption/model? It appears that estimating an adjacent matrix is of central importance. How to interpret it in Fig. 1c? What's the horizontal/vertical dimension? (Are those grid points? if so, how many?) Is it judged by the density of dots? What parameters the MCMC method is used to train? The algorithm is opaque to readers in atmospheric sciences in the present manuscript. It should be much improved for publication. I also suggest giving some examples (some specific ERE events) for illustrative purposes. The ERE index is the outcome of this method. It also needs to be clarified.*

We appreciate the reviewer's input and apologize for the lack of clarity in explaining the methodology. We agree that the climate network approach may not be widely known. Yet, we would like to mention that the climate network method, although not extensively utilized, is an established tool in the field of non-linear time series analysis for meteorology data. It offers valuable insights into the complex interactions and patterns within climatic systems (Dijkstra et al., 2019; Boers et al., 2021). To analyze the synchronization of rainfall extremes the combination of event synchronization and climate networks turned out to be a robust indicator for long-range teleconnections (Boers et al., 2019). Several studies have applied it to analyze spatial patterns of extreme rainfall events during the South Asian Summer Monsoon (Malik et al., 2010; Stolbova et al., 2016; Wolf et al., 2021; Gupta et al., 2022). We agree that improve-

ments are necessary to enhance the clarity of our approach, especially considering that it combines two different methods, namely climate networks and community detection. We have therefore rewritten most parts of the methodology on community detection and the climate network construction. The reviewer is correct in that the schematic in Fig. 1 is essential for understanding the applied methodology but was not intuitive to comprehend. We have therefore updated the schematic and tried to provide better context in it. We further agree that the community-specific synchronous ERE index is an outcome of this method and, thus, we have included it in Fig. 1. Further, we simplified the definition of the index (see eq. 4 in section Community-specific synchronous ERE index) to make it more intuitive to understand.

We would like to briefly answer the specific concerns point-by-point:

- The decision to apply the climate network approach in our study stems from the need to overcome the limitations of traditional time series analysis in meteorology, which primarily focuses on mean and standard deviation calculations. Given the intricate and nonlinear dependencies inherent in climatic systems, the climate network approach, derived from complex network science, offers a valuable extension to traditional methods.
- The network is constructed by employing the event synchronization technique which is a point-wise similarity measure for extreme event time series. The constructed network should not be confused with an artificial neural network and no cost function is used here. One key element is the adjacency matrix A , a square matrix of size $N \times N$, where N represents the number of spatial locations, equivalent to the number of time series under consideration. If $A_{i,j} = 1$ (denoted by a black dot in Fig. 1c), it indicates that events at location i are statistically significantly followed by events at location j (see Methods, sec. 'Constructing the network'). To aid in the interpretation of the adjacency matrix, we have included a further schematic graphic in SI Fig. S3.
- Identifying communities in a network is a widely investigated topic in complex network science and hence, for this purpose, various algorithms have been developed. In this study, we used the Stochastic Block Model (SBM), arguably one of the most established community detection algorithm tools Fortunato and Hric (2016). This algorithm sorts the rows in the adjacency matrix to identify a block structure. We apologize that this reordering was not explained and thus the block structure of the adjacency matrix in Fig. 1c could have been confusing. This is now corrected (see caption Fig. 1 c,d and Methods, sec. 'Estimating communities within climate networks').
- Employing the SBM as a probabilistic generative model, the network communities are inferred using an agglomerative multilevel Markov chain Monte Carlo (MCMC) sampling approach. Initially, node assignments to communities are randomized, and the quality of this assignment is evaluated. Optimization is guided by the Minimum Description Length (MDL) principle, which favors simpler models with equal explanatory power. This is now explained in Methods sec. 'Estimating communities within climate networks'. The applied algorithm combines probabilistic modeling and optimization methods and thus does not necessitate hyperparameters (Peixoto, 2014). The only fixed constraint is the maximum number of groups, which is set to 10, to focus on large-scale structures.

We agree with the reviewer's suggestion of including specific examples of illustrating the

concept of propagating EREs and have included an example of BSISO-related Canonical ERE propagation (SI V.1, Fig. S10,S11) over the Indian Ocean, South Asia, and India towards the Western Pacific for the year 1990. We thus added the following sentence to the main text: “*The combined movement of the eastward and northward propagation characterizes the “canonical” BSISO propagation: A dominant low-frequency mode (30-60 days) in the form of a deep convection zone carrying heavy rainfall emerges in the equatorial Indian Ocean and moves simultaneously eastward and northward, forming a northwest-southeast tilted convection band which, after transgressing the Maritime Continent barrier, progresses further to the Pacific Ocean (exemplified in Fig. S10 and Fig. S11)*”.

2. *The authors study the relationship between the BSISO diversity and ENSO. On Page 4, Fig. 4, and Fig. 9, the authors refer to the third mode in Fig. 4 as “stationary”. There is some indication that this mode has some convective signals in the western Pacific (the plotting domain only goes to 140E, please expand this further east, as Fig. 9). Fig. 9 also shows that there are substantial signals in the western Pacific region after 20 days, indicating eastward propagation (even though it appears not continuous). It appears that this is not stationary mode, but jumps across the MC region and reboots in the western Pacific.*

We agree that including the Western Pacific region until the dateline was lacking for a more comprehensive analysis. We have therefore expanded the plotting domain to include the range until the dateline in Fig. 9. However, we would like to clarify that we believe that the signals observed in the Western Pacific region are not linked to the Stationary mode (as labeled in Fig. 4). We believe that these signals in the Western Pacific are primarily influenced by the anomalous Walker circulation (SI Fig. S28) and the uprising air masses around the dateline, which persists throughout the analysis period (see new Fig. 4c). The impression that the anomalies in the Western Pacific are only present after 20 days may be due to the overlapping of later days in the plot, and we apologize for any confusion caused. Nonetheless, we have carefully examined the data and can confirm that the signals in the Western Pacific region are consistently present throughout the analyzed period. Therefore, we would like to retain the labeling of the mode as “stationary” as we feel that we do not observe a ‘jump’ here.

3. *The authors attribute northward propagation of the BSISO to vertical wind shear. But this is not the only mechanism. Recent observation analysis (Jiang et al., 2018) and theory (Wang and Sobel, 2022) further show that background moisture is crucial for the propagation of the BSISO in the Indian Ocean, where seasonal mean column moisture increases northward and eastward. The theoretical study shows that the BSISO can have both eastward and poleward propagation, and the phase tilts northwest-southeastward in the presence of a poleward gradient of moisture, both consistent with the observed phase tilt of the BSISO in the northern hemisphere. Given these prior studies emphasizing the important roles of moisture for the BSISO, I suggest the authors also examine moisture in different ENSO years (El Niño, La Niña, etc.) to see if the moisture gradients are significantly different in these years, leading to different BSISO modes, i.e., propagation, or blocking, or jumping.*

We sincerely appreciate the reviewer for addressing this crucial point, which helped us

substantially improve our understanding of the physical propagation mechanism. Their valuable comment has led to the development of a new proposal that aims to explain the observed propagation diversity. Based on these new insights, we have also added a new schematic as Fig. 7 describing the propagation mechanism in terms of differences in the east- and westward background MSE gradients to provide the reader some intuition on the differences between the three propagation modes. We fully agree with the reviewer that not incorporating insights of the moisture mode theory in our study presented a significant gap and we also want to thank the reviewer for pointing us towards the relevant work of Jiang et al. (2018) and Wang and Sobel (2022). In the revised version of our manuscript, we have included a comprehensive discussion of the moisture mode theory in the introduction section and our analysis now takes into account the implications and insights offered by this theory. We now refer to the moisture mode theory framework Sobel and Maloney (2012); Kim et al. (2014); Adames and Kim (2016); Wang and Sobel (2022). We also want to thank the reviewer for their proposal to examine the moisture background state for the different propagation modes (Canonical, Blocked or Stationary). We have adopted this proposal in the new Fig. 5 b,d,f by analyzing the background moist static energy (MSE) gradient. These states are modulated by the different ENSO conditions (Fig. 5 a,c,e). This modulation manifests itself through changes in the local overturning circulation. We indeed find differences in the zonal and meridional column MSE gradient. Notably, only the canonical mode shows the expected increases northward and eastward, while the Eastward Blocked type does only show a poleward gradient, and the Stationary neither a poleward nor an eastward increasing gradient. We thus explain the propagation in the revised version as follows: *“Taking together the above results, we propose the following mechanism for the three BSISO propagation modes, schematically visualized in Fig. 7. Differences in the ENSO condition in the tropical Pacific induce changes in the background MSE state over the Maritime Continent via modulation of Walker circulation. In all three modes intensified convection is observed in the equatorial Indian Ocean. For the Canonical mode (Fig. 7 a) the MSE background condition has a zonal gradient over the Maritime Continent (Fig. 5 a) which is induced by the Walker circulation. For the Eastward Blocked mode (Fig. 7 b) the La Niña-like conditions trigger enhanced convection over the Maritime Continent (SI Fig. S33 b) and anomalously wet conditions. Consequently, the presence of easterly winds coming from the tropical Pacific opposing the eastward propagation of the BSISO convective system at the Maritime continent potentially contributes to the observed two opposing MSE gradients preventing the propagation in zonal direction (Fig. 5 d). The northward MSE gradient component remains unaffected. Contrary, for the Stationary mode (Fig. 7 c) the El Niño-like conditions induce a suppression of convection through the reduced low-level winds over the tropical Pacific leading to anomalously dry conditions over the Maritime Continent. The zonal MSE gradient is therefore reduced and the convective system remains over the initiation region in the Equatorial Indian Ocean.”*

Specific comments:

1. *P1. Abstract. The authors emphasize they focus on the region of the South Asian Summer Monsoon. Nevertheless, the region the authors study extends far beyond SASM (e.g., Fig. 2a). It covers most of East Asia, the Maritime Continent, and the vast*

western Pacific Ocean. I would suggest using another name.

We appreciate the reviewer's comment regarding the potential misinterpretation of the term "South Asian Summer Monsoon" as solely focusing on the Mainland Summer Monsoon, which was not the intended scope of our study. To address this concern, we suggest the following terminology: "*Indo-Pacific Asia*". Accordingly, we also changed the title to "*Propagation pathways of Indo-Pacific rainfall extremes.*"

2. *Fig.3, what is the null model? Why it is by construction 10%?*

We agree with the reviewer that the description of the null model was not easy to understand. A null model for Fig. 3 is a random model where random days in JJAS are selected as days of high synchronization $P(EREs = 1)$. We have added a better explanation to the current version of the manuscript as follows: "*As we define days of high synchronization within a community as the top 10% of the community-specific synchronous ERE index, by definition at most 10% of days in JJAS are days of maximum synchronization (dashed lines in Fig. 3). The corresponding null model for a day being a day of high synchronization ($P(EREs=1)$) is therefore by construction 10 %.*"

3. *P4. Is the ERE index used to compute the lag correlation in Figure 2c? The ERE index needs to be shown and checked/compared with other BSISO indexes.*

The reviewer is right that the ERE index was not properly introduced and discussed. The lead-lag correlation plots were computed using the community-specific ERE index of each community and also to define the most synchronous days used to create Fig. 3. We apologize that we did not make the comparison to the BSISO indices. Showing a time series of around 4200 points in time is difficult to visualize meaningfully. We, therefore, compute the correlation matrix of the community-specific ERE indices with the two components of the BSISO index by Kikuchi et al. (2012) and Lee et al. (2013) (SI Fig. S16, S17). We also created a linear model that computes the explained variances of the community-specific ERE indices by the first two components of the BSISO index (SI VII). In the main text, we have added the following phrase in the results section: "*The dependency of the occurrence of EREs and the BSISO can also be shown by a correlation analysis of the community-specific ERE indices to the BSISO index.*"

4. *P6, Figure 4, the unit for OLR anomalies is W/m^2 . Figure 4 shows that OLR anomalies are in the range of $-60 - 60 \cdot 10^3 W/m^2$. This is a huge number for OLR and it is not in a reasonable range.*

Thank you for bringing this discrepancy to our attention. In Fig. 4 the unit was incorrectly stated as W/m^2 instead of J/m^2 . We have rectified this mistake by dividing the values of unit J/m^2 by the accumulation period (1day=3600s) to W/m^2 . The corresponding Fig. 4 is updated. .

5. *P11, Figure 8 shows mean vertical shear, dw/dy , and relative vorticity. The latter two variables look like the BSISO structure. That is, it's a result of BSISO, not the cause. Thus it is hard to use this to explain the different propagation of the BSISO modes.*

Thank you for pointing that out. We agree that these plots cannot be used to explain the different propagation modes. We have omitted the plot in the main text and have moved it to the SI where we describe it as a result of the BSISO that can be also

observed in these variables.

6. *P13, 2nd paragraph in the Data section. Synchronicity: not defined, delete “according to”*

Thank you, we corrected the typo and omitted the term synchronicity here (as it is defined later in the data section).

7. *P13: The BSISO index used in the manuscript has difficulty capturing northward propagation according to lag correlation analysis (see reference 50 (i.e. Wang et al. (2018) author’s note). Please use another one in ref 48 (i.e. Kikuchi et al. (2012)) or 49 (i.e. Kiladis et al. (2014)).*

We thank the reviewer for bringing to our attention that the BSISO index by Kikuchi et al. (2012) better captures the northward propagation compared to the index by Lee et al. (2013). We have repeated the analysis and introduced a new plot in Fig. 3 by using the index by Kikuchi et al. (2012). We further changed the data section in the manuscript accordingly. Note, however, that differences to our previous results are minor and the qualitative conclusions remain unchanged. We introduced a new subsection in the SI to discuss the utilized BSISO index (see SI V, Fig. S15).

8. *P15, 3rd paragraph: “OLR is used instead of precipitation because it directly indicates deep convection in the tropics.” This is not a valid argument. OLR is an indicator of convection, but it’s no better than precipitation. A better argument could be that OLR is a lot smoother than precipitation, and OLR is a direct measurement and it does not need an inverse algorithm (which could have errors) like precipitation.*

We thank the reviewer for putting this right. We agree and have reformulated the phrase by adopting the reviewer’s proposal to: ”*OLR is like precipitation an indicator for deep convective activity. Compared to precipitation, however, OLR exhibits a considerably smoother pattern and offers the advantage of direct measurement without requiring an inverse algorithm (which can introduce errors) as in the case of precipitation.*”

9. *P15, last paragraph: where the Helmholtz decomposition is applied? Global domain or only tropics?*

The Helmholtz decomposition is applied globally on the (u, v) wind fields. This is now also more explicitly stated in the manuscript.

10. *P13, Data section: The MSWEP dataset is a derived product. It needs to be calibrated with surface observation. Do you have a reference for this?*

The reviewer’s point is valid regarding the need for calibration of the MSWEP dataset using surface observations. We acknowledge the importance of ensuring accurate calibration and evaluation to establish reliable measures. We have identified literature that addresses the calibration of the MSWEP dataset especially in the Indian and South East Asian monsoon domain. We have added the following sentence to our data description: “*We use daily precipitation sums for the time period of 1979–2020 from the Multi-Source Weighted-Ensemble Precipitation (MSWEP) V2.2 dataset that merges gauge, satellite (CMORPH, GSMaP-MVK, and TRMM-based), and reanalysis data (ERA5) provided in a resolution of $0.1^\circ \times 0.1^\circ$ (Beck et al., 2019b). We used this dataset as it covers a longer time range than other available multi-satellite precipitation*

products and represents rainfall extremes well on global scales (Beck et al., 2019a) and locally over India (Ali and Mishra, 2018; Prakash, 2019) and South East Asia (Wu and Zhao, 2022; Du et al., 2022)."

Additionally, to validate that our findings are independent of the chosen dataset, we performed a parallel analysis using the TRMM rainfall dataset (Huffman et al., 2007). Remarkably, despite the dataset change, we observed consistent community structures and comparable statistical results in reproducing the BSISO patterns (see SI Fig. S18). This further reinforces the robustness and reliability of our findings.

11. *Figure S1 caption: "An Extreme Rainfall Event (ERE) is defined as a day with more than 1 mm/day of precipitation." This doesn't sound right.*

Thank you, indeed this was a typo. 1 mm/day refers to the definition of a wet day, not an extreme event. We have corrected the typo.

Point-by-point responses to reviewer #3

This article examines South Asian extreme rainfall and propagation of the boreal summer intraseasonal oscillation (BSISO). Exploiting machine learning, spatial patterns, and propagation pattern of precipitation are identified. The propagation mechanisms and the role of ENSO are explored. I find the results inspiring, but I suppose it is not acceptable until the following comments are addressed.

We thank the reviewer for their time and the valuable comments on our manuscript that have helped us refine our analysis. We have adapted the reviewer's comments in the revised version of the manuscript. Please find below our answers to the major and minor comments.

Major comments:

1. *Data. Could you only use data in boreal summer? For two reasons. (1) The study seeks to address South Asian extreme rainfall. (2) Community "BoB" bends back in the Southern Hemisphere, which I assume is dominated by MJOs in the boreal winter. The results (Figures 5-8) could be cleaner if you focus on the boreal summer. The results are inspiring, but I would love to hold back a bit till the authors eliminate seasonality in the analysis.*

We believe that this might be a misunderstanding. We did already use only data in the boreal summer season from June through September (JJAS) for exactly the reason the reviewer points out. We apologize that we did not make this clear enough in the submitted version of the manuscript. The results section now starts with: "*We thus identify geographical regions where EREs (defined locally as days with rainfall sums above the 90th percentile of wet days) occur synchronously (within up to 10 days) on average over the boreal summer JJAS data period.*"

2. *The first three figures are disconnected from the rest of the figures. Community detection is a novel method to study the BSISO and indeed, it yields a similar pattern as that from the more conventional EOF analysis. But as I thought the authors would complete the story in the newly defined "community" space, the rest of the article returns to conventional K-mean clustering in the conventional Hovmöller diagram. I am not suggesting conventional methods are worse choices – I just do not understand why community detection is necessary for this study.*

We appreciate the feedback that the motivation for why the community detection approach was necessary was not clear enough. While we understand the reviewer's perspective, we would like to emphasize that we see both community detection and K-means clustering as serving their appropriate roles in our analysis. Community detection plays a crucial role in identifying relevant regions, defining potential BSISO propagation events, and examining the propagation of extremes by connecting those to the BSISO. Thus, it allows us to capture the spatial organization and interdependencies among regions in terms of extreme rainfall behavior. On the other hand, K-means clustering is a powerful tool when it comes to grouping patterns, especially in two-dimensional images such as those shown in Fig 4. It enables us to uncover distinct clusters and reveal underlying patterns in the data. Therefore, we believe that the integration of community detection and K-means clustering offers a more comprehensive and nuanced analysis of the dataset.

To better link the first subsection to the next subsections, we added the following paragraph at the beginning of the second subsection: “*The preceding subsection has demonstrated that the BSISO plays a crucial role in shaping the spatial distribution of extreme rainfalls and provides insights into potential propagation pathways (Fig. 2 b). However, the signal from the BSISO in the Equatorial Indian Ocean towards the Western Pacific weakens over time (Fig. 2 c-f). Therefore, we investigate potential drivers of this diversity in propagation.*”

Minor comments:

1. *How is the number of communities determined? Is it sensitive to the chosen community detection algorithm?*

We thank the reviewer for addressing this important point and acknowledge that the clarity of this information was lacking in the initial version of the manuscript. The determination of the number of communities is a result of the employed community detection algorithm. It aims to optimize the identification of communities that best describes the network structure while minimizing the number of communities. We have extensively revised the methods section pertaining to community detection, aiming to provide a more comprehensive explanation. For further elaboration, we kindly refer the reviewer to our response to general comment 2 of Reviewer #2 above. Further, in the updated version of the manuscript we have included in the SI a comparison to a community detection approach using the Parallel Louvain Method (PLM), implemented in the NetworkIT packages (Staudt et al., 2014). The qualitative structure of the identified communities turned out to be similar also to other community detection algorithms with some minor differences regarding the BoB and the EIO community. We have chosen the SBM implementation approach, as it turned out to provide more stable results in the spatial shapes of the communities and the membership likelihoods per community were cleaner.

We have added a brief discussion on the dependency on the chosen community detection algorithm in the discussion section of the revised article.

References

- Á. F. Adames and D. Kim. The MJO as a Dispersive, Convectively Coupled Moisture Wave: Theory and Observations. *J. Atmos. Sci.*, 73(3):913–941, Mar. 2016. ISSN 0022-4928. doi: 10.1175/JAS-D-15-0170.1.
- H. Ali and V. Mishra. Increase in Subdaily Precipitation Extremes in India Under 1.5 and 2.0 °C Warming Worlds. *Geophys. Res. Lett.*, 45(14):6972–6982, July 2018. ISSN 0094-8276. doi: 10.1029/2018GL078689.
- H. E. Beck, M. Pan, T. Roy, G. P. Weedon, F. Pappenberger, A. I. J. M. van Dijk, G. J. Huffman, R. F. Adler, and E. F. Wood. Daily evaluation of 26 precipitation datasets using Stage-IV gauge-radar data for the CONUS. *Hydrol. Earth Syst. Sci.*, 23(1):207–224, Jan. 2019a. ISSN 1027-5606. doi: 10.5194/hess-23-207-2019.
- H. E. Beck, E. F. Wood, M. Pan, C. K. Fisher, D. G. Miralles, A. I. J. M. van Dijk, T. R. McVicar, and R. F. Adler. MSWEP V2 Global 3-Hourly 0.1° Precipitation: Methodology and Quantitative Assessment. *Bull. Am. Meteorol. Soc.*, 100(3):473–500, Mar 2019b. ISSN 0003-0007. doi: 10.1175/BAMS-D-17-0138.1.
- N. Boers, B. Goswami, A. Rheinwalt, B. Bookhagen, B. Hoskins, and J. Kurths. Complex networks reveal global pattern of extreme-rainfall teleconnections. *Nature*, 566(7744):373–377, feb 2019. ISSN 0028-0836. doi: 10.1038/s41586-018-0872-x. URL <http://www.nature.com/articles/s41586-018-0872-x>.
- N. Boers, J. Kurths, and N. Marwan. Complex systems approaches for Earth system data analysis. *J. Phys.: Complexity*, 2(1):011001, Apr. 2021. ISSN 2632-072X. doi: 10.1088/2632-072x/abd8db.
- H. A. Dijkstra, E. Hernández-García, C. Masoller, and M. Barreiro. *Networks in Climate*. Cambridge University Press, Cambridge, England, UK, Feb. 2019. ISBN 978-1-31627575-7. doi: 10.1017/9781316275757.
- Y. Du, D. Wang, J. Zhu, Z. Lin, and Y. Zhong. Intercomparison of multiple high-resolution precipitation products over China: Climatology and extremes. *Atmos. Res.*, 278:106342, Nov. 2022. ISSN 0169-8095. doi: 10.1016/j.atmosres.2022.106342.
- S. Fortunato and D. Hric. Community detection in networks: A user guide. *Phys. Rep.*, 659: 1–44, Nov. 2016. ISSN 0370-1573. doi: 10.1016/j.physrep.2016.09.002.
- J. Gao, H. Lin, L. You, and S. Chen. Monitoring early-flood season intraseasonal oscillations and persistent heavy rainfall in South China. *Clim. Dyn.*, 47(12):3845–3861, Dec. 2016. ISSN 1432-0894. doi: 10.1007/s00382-016-3045-3.
- S. Gupta, N. Boers, F. Pappenberger, and J. Kurths. Complex network approach for detecting tropical cyclones. *Clim. Dyn.*, 57(11):3355–3364, Dec. 2021. ISSN 1432-0894. doi: 10.1007/s00382-021-05871-0.
- S. Gupta, Z. Su, N. Boers, J. Kurths, N. Marwan, and F. Pappenberger. Interconnection between the Indian and the East Asian Summer Monsoon: spatial synchronization patterns of extreme rainfall events. *Int. J. Climatol.*, n/a(n/a), Sept. 2022. ISSN 0899-8418. doi: 10.1002/joc.7861.

- G. J. Huffman, R. F. Adler, D. T. Bolvin, G. Gu, E. J. Nelkin, K. P. Bowman, Y. Hong, E. F. Stocker, and D. B. Wolff. The TRMM Multisatellite Precipitation Analysis (TMPA): Quasi-global, multiyear, combined-sensor precipitation estimates at fine scales. *Journal of Hydrometeorology*, 8(1):38–55, 2007. ISSN 1525755X. doi: 10.1175/JHM560.1.
- X. Jiang, Á. F. Adames, M. Zhao, D. Waliser, and E. Maloney. A Unified Moisture Mode Framework for Seasonality of the Madden–Julian Oscillation. *J. Clim.*, 31(11):4215–4224, June 2018. ISSN 0894-8755. doi: 10.1175/JCLI-D-17-0671.1.
- S. Kemball-Cook and B. Wang. Equatorial Waves and Air–Sea Interaction in the Boreal Summer Intraseasonal Oscillation. *J. Clim.*, 14(13):2923–2942, July 2001. ISSN 0894-8755. doi: 10.1175/1520-0442(2001)014<2923:EWAASI>2.0.CO;2.
- K. Kikuchi. The Boreal Summer Intraseasonal Oscillation (BSISO): A Review. *Journal of the Meteorological Society of Japan. Ser. II*, pages 2021–045, Mar. 2021. ISSN 0026-1165. doi: 10.2151/jmsj.2021-045.
- K. Kikuchi, B. Wang, and Y. Kajikawa. Bimodal representation of the tropical intraseasonal oscillation. *Clim. Dyn.*, 38(9):1989–2000, May 2012. ISSN 1432-0894. doi: 10.1007/s00382-011-1159-1.
- G. N. Kiladis, J. Dias, K. H. Straub, M. C. Wheeler, S. N. Tulich, K. Kikuchi, K. M. Weickmann, and M. J. Ventrice. A Comparison of OLR and Circulation-Based Indices for Tracking the MJO. *Mon. Weather Rev.*, 142(5):1697–1715, May 2014. ISSN 1520-0493. doi: 10.1175/MWR-D-13-00301.1.
- D. Kim, J.-S. Kug, and A. H. Sobel. Propagating versus Nonpropagating Madden–Julian Oscillation Events. *J. Clim.*, 27(1):111–125, Jan. 2014. ISSN 0894-8755. doi: 10.1175/JCLI-D-13-00084.1.
- J.-Y. Lee, B. Wang, M. C. Wheeler, X. Fu, D. E. Waliser, and I.-S. Kang. Real-time multivariate indices for the boreal summer intraseasonal oscillation over the Asian summer monsoon region. *Clim. Dyn.*, 40(1):493–509, Jan. 2013. ISSN 1432-0894. doi: 10.1007/s00382-012-1544-4.
- H. Lin. Monitoring and Predicting the Intraseasonal Variability of the East Asian–Western North Pacific Summer Monsoon. *Mon. Weather Rev.*, 141(3):1124–1138, Mar. 2013. ISSN 1520-0493. doi: 10.1175/MWR-D-12-00087.1.
- N. Malik, N. Marwan, and J. Kurths. Spatial structures and directionalities in Monsoonal precipitation over South Asia. *Nonlinear Processes Geophys.*, 17(5):371–381, Sep 2010. ISSN 1023-5809. doi: 10.5194/npg-17-371-2010.
- T. P. Peixoto. Hierarchical block structures and high-resolution model selection in large networks. *Physical Review X*, 4(1):1–18, 2014. ISSN 21603308. doi: 10.1103/PhysRevX.4.011047.
- S. Prakash. Performance assessment of CHIRPS, MSWEP, SM2RAIN-CCI, and TMPA precipitation products across India. *J. Hydrol.*, 571:50–59, Apr. 2019. ISSN 0022-1694. doi: 10.1016/j.jhydrol.2019.01.036.
- A. Rheinwalt, N. Marwan, J. Kurths, P. Werner, and F.-W. Gerstengarbe. Boundary effects in network measures of spatially embedded networks. *Europhys. Lett.*, 100(2):28002, Oct. 2012. ISSN 0295-5075. doi: 10.1209/0295-5075/100/28002.

- J. Schwendike, P. Govekar, M. J. Reeder, R. Wardle, G. J. Berry, and C. Jakob. Local partitioning of the overturning circulation in the tropics and the connection to the Hadley and Walker circulations. *J. Geophys. Res. Atmos.*, 119(3):1322–1339, Feb. 2014. ISSN 2169-897X. doi: 10.1002/2013JD020742.
- D. Sengupta, B. N. Goswami, and R. Senan. Coherent intraseasonal oscillations of ocean and atmosphere during the Asian Summer Monsoon. *Geophys. Res. Lett.*, 28(21):4127–4130, Nov. 2001. ISSN 0094-8276. doi: 10.1029/2001GL013587.
- A. Sobel and E. Maloney. An Idealized Semi-Empirical Framework for Modeling the Madden–Julian Oscillation. *J. Atmos. Sci.*, 69(5):1691–1705, May 2012. ISSN 0022-4928. doi: 10.1175/JAS-D-11-0118.1.
- A. H. Sobel, E. D. Maloney, G. Bellon, and D. M. Frierson. The role of surface heat fluxes in tropical intraseasonal oscillations. *Nat. Geosci.*, 1:653–657, Oct. 2008. ISSN 1752-0908. doi: 10.1038/ngeo312.
- C. L. Staudt, A. Sazonovs, and H. Meyerhenke. NetworKit: A Tool Suite for Large-scale Complex Network Analysis. *arXiv*, Mar. 2014. doi: 10.48550/arXiv.1403.3005.
- V. Stolbova, E. Surovyatkina, B. Bookhagen, and J. Kurths. Tipping elements of the Indian monsoon: Prediction of onset and withdrawal. *Geophys. Res. Lett.*, 43(8):3982–3990, Apr. 2016. ISSN 0094-8276. doi: 10.1002/2016GL068392.
- B. Wang. Intraseasonal Modulation of the Indian Summer Monsoon. In *Oxford Research Encyclopedia of Climate Science*. Oxford University Press, Apr. 2018. doi: 10.1093/acrefore/9780190228620.013.616.
- S. Wang and A. H. Sobel. A Unified Moisture Mode Theory for the Madden–Julian Oscillation and the Boreal Summer Intraseasonal Oscillation. *J. Clim.*, 35(4):1267–1291, Feb. 2022. ISSN 0894-8755. doi: 10.1175/JCLI-D-21-0361.1.
- S. Wang, D. Ma, A. H. Sobel, and M. K. Tippett. Propagation Characteristics of BSISO Indices. *Geophys. Res. Lett.*, 45(18):9934–9943, Sept. 2018. ISSN 0094-8276. doi: 10.1029/2018GL078321.
- M. Wen, T. Li, R. Zhang, and Y. Qi. Structure and Origin of the Quasi-Biweekly Oscillation over the Tropical Indian Ocean in Boreal Spring. *J. Atmos. Sci.*, 67(6):1965–1982, June 2010. ISSN 0022-4928. doi: 10.1175/2009JAS3105.1.
- F. Wolf, U. Ozturk, K. Cheung, and R. V. Donner. Spatiotemporal patterns of synchronous heavy rainfall events in East Asia during the Baiu season. *Earth Syst. Dyn.*, 12(1):295–312, Mar. 2021. ISSN 2190-4979. doi: 10.5194/esd-12-295-2021.
- X. Wu and N. Zhao. Evaluation and Comparison of Six High-Resolution Daily Precipitation Products in Mainland China. *Remote Sens.*, 15(1):223, Dec. 2022. ISSN 2072-4292. doi: 10.3390/rs15010223.

REVIEWERS' COMMENTS

Reviewer #1 (Remarks to the Author):

I appreciate the effort from the authors who have responded to my comments to the original manuscript in details, and have performed extra analyses to reply to the comments.

The authors have performed spectral analysis of the synchronous event indices and confirmed that the Maritime Continent and West Pacific consist more of the high-frequency BISIO mode, while the other communities consist more of the low-frequency mode. This mode separation has been discussed in the Introduction, results and the final discussion section.

The authors have also added discussion according to my comments on earlier studies on MJO propagation (Kemball-Cook and Wang 2001). The authors have examined sub-seasonal variability of their results, which confirmed that the synchronous days would not vary much at least in JJAS. In addition, the authors have examined air-sea interaction processes in the new Fig. 5 and new schematic Fig. 7 (and through downward shortwave radiation, moisture and surface latent heat flux in Fig. S34 and MSE and OLR in Fig. S35). This moisture mode theory consideration is related to a comment by reviewer 2 as well.

I thank the authors for adding the network metrics analysis in Figs. S22 to S28. Except the land-sea contrast, such as for the distinct North India-China community, they do not distinguish the other communities very well. Nevertheless, the metrics are of good reference to other studies that apply the ES technique to construct complex networks.

I recommend that the manuscript can be accepted for publication in its current form.

Reviewer #2 (Remarks to the Author):

The authors have made an effort to revise the manuscript, which has led to much-improved manuscript. I have two additional comments on the revision.

1. The third mode in Fig. 4e and f is propagating within the Indian Ocean, e.g., convective anomaly in panel e propagates slowly from 50-90E from day -5 to day 10, and also slightly poleward propagation in panel f. So, it's not really stationary, but a mode trapped in the Indian Ocean. I would suggest change its name to something else to avoid misunderstanding, e.g., Indian Ocean trapped mode, or quasi-stationary mode.

2. first two paragraphs in Page 12. It's claimed that the responses in Fig. 8 are the Kelvin and Rossby waves (Fig. 8). But it's not clear how the authors identified these as Kelvin/Rossby waves. I don't see the manuscript provide any evidence to support this claim. The left column of Fig. 8 shows zonal wind features near the equator. If they are Kelvin waves, do they propagate at similar speed as dry or convectively coupled Kelvin waves? The right column shows meridional wind near the equator and off equatorial regions. Do they propagate westward like Rossby waves? Or, are these signals propagate at similar speed as the BSISO? If yes, these signals (Fig 8) could be part of the BSISO signals rather than mere responses. I believe it is important to clarify this section and revise this discussion accordingly.

Reviewer #3 (Remarks to the Author):

The authors have almost addressed my comments. I have one minor comment left. I apologize if I didn't make it clear last time – my previous Major Comment 2 is partly about the connection between the “community space” and conventional longitude/latitude space.

To be specific, Figure 2b can be regarded as a Hovmoller diagram in the community space while Figure 4 shows Hovmoller diagrams in the conventional longitude/latitude space. To illustrate coherence, I'd suggest adding two subplots and translate Figure 2b into "conventional" Hovmoller diagrams and translate Figure 4 into Hovmoller diagrams in the community space. Or, as I commented last time, one can conduct K-mean clustering analysis in the community space, as opposed to the longitude/latitude space.

Felix Strnad
University of Tübingen
Cluster of Excellence 'Machine Learning'
Maria-von-Linden-Strasse 6
D-72076 Tübingen
email: felix.strnad@uni-tuebingen.de

Tübingen, August 16, 2023

Response letter for “Propagation pathways of Indo-Pacific rainfall extremes are modulated by Pacific sea surface temperatures”

Point-by-point responses to reviewer #1

I appreciate the effort from the authors who have responded to my comments on the original manuscript in detail and have performed extra analyses to reply to the comments. The authors have performed spectral analysis of the synchronous event indices and confirmed that the Maritime Continent and West Pacific consist more of the high-frequency BISIO mode, while the other communities consist more of the low-frequency mode. This mode separation has been discussed in the Introduction, results and the final discussion section. The authors have also added discussion according to my comments on earlier studies on MJO propagation Kemball-Cook and Wang (2001). The authors have examined sub-seasonal variability of their results, which confirmed that the synchronous days would not vary much at least in JJAS. In addition, the authors have examined air-sea interaction processes in the new Fig. 5 and new schematic Fig. 7 (and through downward shortwave radiation, moisture, and surface latent heat flux in Fig. S34 and MSE and OLR in Fig. S35). This moisture mode theory consideration is related to a comment by reviewer 2 as well. I thank the authors for adding the network metrics analysis in Figs. S22 to S28. Except for the land-sea contrast, such as for the distinct North India-China community, they do not distinguish the other communities very well. Nevertheless, the metrics are of good reference to other studies that apply the ES technique to construct complex networks. I recommend that the manuscript can be accepted for publication in its current form.

We would like to thank the reviewer again for the thorough evaluation of our manuscript and the valuable comments provided in the first round of reviews. We are delighted that our revisions addressed their concerns, and we appreciate the recognition of our efforts to improve the paper and are grateful for the positive evaluation.

Point-by-point responses to reviewer #2

The authors have made an effort to revise the manuscript, which has led to a much-improved manuscript. I have two additional comments on the revision.

We first want to thank the reviewer again for their time and their very helpful and detailed comments. The reviewer's feedback has been very helpful in enhancing the overall quality of the article and, therefore, we are happy that also the reviewer acknowledges these improvements.

Please find below our answers to the two additional comments.

Comments:

1. *The third mode in Fig. 4e and f is propagating within the Indian Ocean, e.g., a convective anomaly in panel e propagates slowly from 50-90E from day -5 to day 10, and also slightly poleward propagation in panel f. So, it's not really stationary, but a mode trapped in the Indian Ocean. I would suggest changing its name to something else to avoid misunderstanding, e.g., Indian Ocean trapped mode, or quasi-stationary mode.*

We fully agree with the observation regarding the third mode in Fig. 4 e, f, and we acknowledge that it indeed exhibits propagation within the Indian Ocean, and the labeling as 'stationary' might be misleading. Therefore, we thank the reviewer for the suggestion for the term "quasi-stationary mode". We have now adopted the term "quasi-stationary mode" in the current version of the manuscript to better describe this specific mode. Accordingly, we have updated all figures in the main text. The following phrases have been added to describe the quasi-stationary mode: "*This mode, consisting of 27 samples, shows a slow propagation that is, however, constraint to the Indian Ocean from 50°E–90°E (Fig. 4 e), and a slightly poleward propagation (Fig. 4 f) from day -5 to day 10.*".

2. *First two paragraphs on Page 12. It's claimed that the responses in Fig. 8 are the Kelvin and Rossby waves (Fig. 8). But it's not clear how the authors identified these as Kelvin/Rossby waves. I don't see the manuscript provide any evidence to support this claim. The left column of Fig. 8 shows zonal wind features near the equator. If they are Kelvin waves, do they propagate at a similar speed as dry or convectively coupled Kelvin waves? The right column shows meridional wind near the equator and off equatorial regions. Do they propagate westward like Rossby waves? Or, are these signals propagate at a similar speed as the BSISO? If yes, these signals (Fig 8) could be part of the BSISO signals rather than mere responses. I believe it is important to clarify this section and revise this discussion accordingly.*

We apologize for the lack of clarity in the description of Fig. 8. In the revised version of the manuscript, this subsection has been rewritten and in particular, a better link to current literature has been provided. The reviewer makes a reasonable point that we did not give evidence that these features in Fig. 8 show propagation characteristics of a Kelvin wave or Rossby wave, rather than just bearing resemblance. To address this concern, we have thus incorporated Hovmöller diagrams of low-level meridional V-winds at 850 hPa. These plots (Fig. 8 b,d,f) show better the westward-traveling nature of the wave.

However, our intention was not to examine the wave characteristics in detail but rather to show that our reported different propagation modes show characteristic features of Kelvin and Rossby wave responses as they have been described in previous literature. In the revised version of the manuscript we therefore refrained from using the term “*Kelvin wave responses*” and use the term “*Kelvin (Rossby)-wave-like patterns*” instead. Further, we now take care to refer the reader to previous literature that describes similar patterns that we also observe in our analysis with respect to the Kelvin- and Rossby waves. For example, the wind field in Fig. 8 a,d resembles a Kelvin wave signature as it has been described in e.g. (Adames and Kim, 2016), Fig.14, (Wang et al., 2019), Fig.4 and (Vallis, 2021), Fig.12). The Rossby wave pattern in the form of Hovmöller diagrams has been presented for example already in Wang and Xie (1997), Fig.6). Further, our intention was to demonstrate the consistency for the Quasi-stationary mode by showcasing that these characteristic Rossby- and Kelvin-wave patterns are not observed here. To address the concern regarding the propagation direction, we have included the zonal wind component as a third column. To address the reviewer’s concerns we added respectively corrected the following paragraph to the description of Fig. 8: “*We observe the characteristic zonally Rossby wave pattern in the low-level winds (Wang and Xie, 1997; Wang et al., 2005; Jiang et al., 2018; Adames and Kim, 2016) initiated at the Maritime Continent at $\sim 120^{\circ}\text{E}$ most clearly for the Canonical mode in Fig. 8 b. The meridional wind anomalies reveal a westward drift in northeast-southwest tilted bands. The westward-oriented waves occur 10 to 20 days after initiation consistent with the propagation of EREs (compare Fig. 2 b). According to previous literature (Wang and Xie, 1997; Adames and Kim, 2016), the anomalous BSISO circulation near the equator exhibits modified Gill-type responses with a stronger amplitude to the north than to the south of the equator. These are associated with the slanted northwest-southeast BSISO rainfall anomalies near the equator and suppressed convection north of 10°N that we also observe in Fig. 8 a,c.*”

Point-by-point responses to reviewer #3

The authors have almost addressed my comments. I have one minor comment left. I apologize if I didn't make it clear last time – my previous Major Comment 2 is partly about the connection between the “community space” and conventional longitude/latitude space.

We thank the reviewer again for their time and the valuable comments and are happy to hear that almost all comments are addressed. We appreciate the reviewer's detailed clarification and apologize again for our previous lack of understanding. We believe that we now better understood the reviewer's point which helped us to refine our analysis. We have included the reviewer's comments in the revised version of the manuscript. Please find below our detailed answer.

Comment:

1. *To be specific, Figure 2b can be regarded as a Hovmoller diagram in the community space while Figure 4 shows Hovmoller diagrams in the conventional longitude/latitude space. To illustrate coherence, I'd suggest adding two subplots and translate Figure 2b into “conventional” Hovmoller diagrams and translate Figure 4 into Hovmoller diagrams in the community space. Or, as I commented last time, one can conduct K-means clustering analysis in the community space, as opposed to the longitude/latitude space. We fully agree with the reviewer that incorporating the Hovmöller diagrams of zonal and meridional OLR would have helped to better distinguish between the physical (lon-lat) space and the community space. It is indeed important to ensure consistency in our narrative, particularly in combination with Fig. 4. The respective Hovmöller diagrams are now shown in SI Fig.S35. We still believe that we need to keep the current Fig. 2 b as it allows conclusions on the extreme rainfall propagation and want to consider that the complexity of Fig. 2 is already very high. We therefore would rather like to refer to the Hovmöller diagrams in the SI. In the main text in the manuscript we added the following clarification to the analysis of Fig. 2: “The composited Hovmöller diagrams for the initiation time points show discernible patterns of eastward propagation through the Maritime Continent and simultaneous northward propagation, but nevertheless, the clarity of the propagation pattern diminishes approximately 5-10 days post initiation, indicating a degree of variability within the propagation pathways (Fig. S8). We thus use the single diagrams for each initiation point in time as input samples to a K-means clustering algorithm (see Methods).”*

The reviewer's suggestion regarding K-means clustering in the community space is very valuable. While we attempted this approach, the outcomes lacked meaningful interpretations. Our analysis implies that the granularity of the five communities in the spatial dimension might be too coarse to yield informative results in the community space. We believe the Hovmöller diagrams in the physical space offer more insightful spatio-temporal features. We created however Hovmöller diagrams in the community space of the results on clustering (in physical space) of BSISO propagation events showcasing distinct propagation modes in the community space presented in SI Fig. S15 These plots are referred to when elaborating on the potential for early warning signals: “These three propagation modes also translate to propagation of EREs via the identified communities (Fig. 2 a), presented in Fig. S15. Using these different propagation modes, it is therefore justified to explore the possibility of Early-Warning signals (EWS) for EREs that are driven by the BSISO at a time horizon of multiple weeks.”

References

- Á. F. Adames and D. Kim. The MJO as a Dispersive, Convectively Coupled Moisture Wave: Theory and Observations. *J. Atmos. Sci.*, 73(3):913–941, Mar. 2016. ISSN 0022-4928. doi: 10.1175/JAS-D-15-0170.1.
- X. Jiang, Á. F. Adames, M. Zhao, D. Waliser, and E. Maloney. A Unified Moisture Mode Framework for Seasonality of the Madden–Julian Oscillation. *J. Clim.*, 31(11):4215–4224, June 2018. ISSN 0894-8755. doi: 10.1175/JCLI-D-17-0671.1.
- S. Kemball-Cook and B. Wang. Equatorial Waves and Air–Sea Interaction in the Boreal Summer Intraseasonal Oscillation. *J. Clim.*, 14(13):2923–2942, July 2001. ISSN 0894-8755. doi: 10.1175/1520-0442(2001)014<2923:EWAASI>2.0.CO;2.
- G. K. Vallis. Distilling the mechanism for the Madden–Julian Oscillation into a simple translating structure. *Q. J. R. Meteorolog. Soc.*, 147(738):3032–3047, July 2021. ISSN 0035-9009. doi: 10.1002/qj.4114.
- B. Wang and X. Xie. A Model for the Boreal Summer Intraseasonal Oscillation. *J. Atmos. Sci.*, 54(1):72–86, Jan. 1997. ISSN 0022-4928. doi: 10.1175/1520-0469(1997)054<0072:AMFTBS>2.0.CO;2.
- B. Wang, P. J. Webster, and H. Teng. Antecedents and self-induction of active-break south Asian monsoon unraveled by satellites. *Geophys. Res. Lett.*, 32(4), Feb. 2005. ISSN 0094-8276. doi: 10.1029/2004GL020996.
- S. Wang, A. H. Sobel, M. K. Tippett, and F. Vitart. Prediction and predictability of tropical intraseasonal convection: seasonal dependence and the Maritime Continent prediction barrier. *Clim. Dyn.*, 52(9):6015–6031, May 2019. ISSN 1432-0894. doi: 10.1007/s00382-018-4492-9.